EMBO
Molecular Medicine

# Biofabricating murine and human myo-substitutes for rapid volumetric muscle loss restoration

Marco Costantini[1,†], Stefano Testa[2,†], Ersilia Fornetti[2], Claudia Fuoco[2], Carles Sanchez Riera[2], Minghao Nie[3], Sergio Bernardini[2], Alberto Rainer[4,5], Jacopo Baldi[6], Carmine Zoccali[6], Roberto Biagini[6], Luisa Castagnoli[2], Libero Vitiello[7], Bert Blaauw[8], Dror Seliktar[9], Wojciech Święszkowski[10], Piotr Garstecki[1], Shoji Takeuchi[3,11], Gianni Cesareni[2,12] , Stefano Cannata[2] & Cesare Gargioli[2,*]

## Abstract

The importance of skeletal muscle tissue is undoubted being the controller of several vital functions including respiration and all voluntary locomotion activities. However, its regenerative capability is limited and significant tissue loss often leads to a chronic pathologic condition known as volumetric muscle loss. Here, we propose a biofabrication approach to rapidly restore skeletal muscle mass, 3D histoarchitecture, and functionality. By recapitulating muscle anisotropic organization at the microscale level, we demonstrate to efficiently guide cell differentiation and myobundle formation both *in vitro* and *in vivo*. Of note, upon implantation, the biofabricated myo-substitutes support the formation of new blood vessels and neuromuscular junctions—pivotal aspects for cell survival and muscle contractile functionalities—together with an advanced muscle mass and force recovery. Altogether, these data represent a solid base for further testing the myo-substitutes in large animal size and a promising platform to be eventually translated into clinical scenarios.

**Keywords** bioprinting; skeletal muscle; stem cells; tissue engineering; VML
**Subject Categories** Methods & Resources; Musculoskeletal System

## Introduction

Skeletal muscle (SM) tissue accounts for over 40% of the total body mass, controlling all voluntary locomotion activities and regulating vital functions such as respiration and metabolic homeostasis. Its functionality is therefore essential to maintain an adequate life quality (Carlson, 1973; Huard *et al*, 2002; Chargé & Rudnicki, 2004; Rizzi *et al*, 2012).

The clinical scenarios of SM-related disorders are complex as the functionality of this tissue can be impaired by several conditions that can either be acquired, as a consequence of contusions, lacerations, or surgical interventions (sarcoma removal), or be congenital/genetic as in muscular dystrophies or metabolic diseases. Regardless of the origin, treating any type of SM loss or disorder remains an unmet therapeutic need (Grogan & Hsu, 2011; Mercuri & Muntoni, 2013).

In this context, skeletal muscle tissue engineering (often referred as SMTE) holds great promises for addressing the two key challenges in the field: (i) the development of reliable 3D *in vitro* models, and (ii) the fabrication of a macroscopic engineered SM tissue (construct volume > 1 cm$^3$; Costantini *et al*, 2017; Fuoco *et al*, 2016a,b).

As far as the first challenge is concerned, SMTE is expected to yield in the near future functional *in vitro* models for developing and testing approaches that may help addressing muscle dysfunctions caused by genetic or metabolic diseases. The success in the development of dependable *in vitro* models of human SM, aside

---

1 Institute of Physical Chemistry, Polish Academy of Sciences, Warsaw, Poland
2 Department of Biology, Rome University Tor Vergata, Rome, Italy
3 Department of Mechano-Informatics, Graduate School of Information Science and Technology, The University of Tokyo, Tokyo, Japan
4 Department of Engineering, Università Campus Bio-Medico di Roma, Rome, Italy
5 Institute of Nanotechnology (NANOTEC), National Research Council, Lecce, Italy
6 IRCCS Regina Elena National Cancer Institute, Rome, Italy
7 Department of Biology, University of Padova, Padova, Italy
8 Department of Biomedical Science and Venetian Institute of Molecular Medicine, University of Padova, Padova, Italy
9 Department of Biomedical Engineering, Techion Institute, Haifa, Israel
10 Faculty of Materials Science and Engineering, Warsaw University of Technology, Warsaw, Poland
11 Institute of Industrial Science, The University of Tokyo, Tokyo, Japan
12 IRCCS Fondazione Santa Lucia, Rome, Italy
*Corresponding author. Tel: +39 06 72594815; E-mail: cesare.gargioli@uniroma2.it
†These authors contributed equally to this work

from contributing to sparing animal experimentation, is expected to overcome some of the problems caused by the failure of some animal models and to reproduce some important features of the human disease. In addition, such *in vitro* models will therefore be of great help in better understanding the etiology of SM-specific diseases—including dystrophies such as sarcoglycanopathy or facio-scapulo-humeral systrophy and metabolic disorders such as Pompe disease and mitochondrial metabolic myopathy (Tanaka *et al*, 2013; Caron *et al*, 2016; Chal *et al*, 2016). Finally, in the context of drug discovery and development, SMTE-based *in vitro* models will be especially valuable for the characterization of drug molecular mechanisms, rapid prioritization of lead candidates, toxicity testing, and new biomarker identification (Smith *et al*, 2016).

The development of *in vitro* SM tissue models has been at the core of intensive industrial and academic research over the past ten years, and an ever-increasing number of innovative approaches have been proposed including 3D bioprinting and organ-on-a chip solutions (Truskey *et al*, 2013; Juhas *et al*, 2016; Agrawal *et al*, 2017; Ostrovidov *et al*, 2019).

Despite the introduction of new technologies and refined protocols for culturing engineered muscles—derived from either murine or human muscle progenitors—so far, SM maturation and maintenance have been only partially recapitulated *in vitro* and, most importantly, to a small scale (Leng *et al*, 2012; Zhang *et al*, 2015). These approaches, while promising, are still far from satisfactorily addressing the second key challenge of SMTE—i.e., the development of macroscopic tissue equivalents of a size suitable for treating volumetric muscle loss (VML): a condition resulting from large traumatic injury or massive surgical ablation upon tumor removal (Passipieri & Christ, 2016; Gilbert-Honick & Grayson, 2019). VML, which is characterized by muscular mass loss, functional deficit, and permanent disability, is currently treated by surgical tissue transfer (autologous transplantation). This invasive approach is often associated with unsatisfactory outcomes due to poor tissue engraftment and donor site morbidity (Khouri *et al*, 1998; Lin *et al*, 1999; Machingal *et al*, 2011).

In this study, we present a microfluidic wet-spinning system (Rinoldi *et al*, 2019) that allows rapid biofabrication (fiber extrusion velocity ≈ 4 m/min) of macroscopic yarns of cell-laden hydrogel microfibers—loaded either with murine (mesoangioblasts, Mabs) or with human (primary myoblasts) muscle precursors—that closely mimic the structure of the SM tissue. Our findings demonstrate that by optimizing key parameters such as: (i) hydrogel precursor composition (i.e., bioink), (ii) cell seeding density, (iii) fiber architecture, and (iv) culturing protocol, we are able to successfully fabricate advanced artificial myo-substitutes that can be used to restore SM mass loss and functionality *in vivo*.

# Results

### Assembly and characterization of a wet-spinning setup

Nowadays, biofabrication technologies hold great promises for the *in vitro* production of functional tissue equivalents thanks to their flexibility and unprecedented precision in cell and biomaterial deposition. Such features are of paramount importance for the recapitulation of organ/tissue complexity and functionality, especially in the

case of skeletal muscle, a tissue characterized by a hierarchical, uniaxially aligned architecture (Frontera & Ochala, 2015).

In a recent study, we presented an extrusion-based 3D bioprinting approach that enables to biofabricate hydrogel scaffolds composed of aligned myoblast-laden fibers (Costantini *et al*, 2017). Despite the remarkable results in terms of muscle architectural guidance, as shown by the accomplishment of oriented, mature myofibers that spontaneously contract *in vitro*, the developed approach suffers from several limitations. Firstly, ectopic implantation in a mouse model revealed some limitations in terms of scaffold remodeling, as large areas were actually devoid of myotubes (likely because of the large diameter of printed fibers ≈ 250 μm). Furthermore, in order to carefully pack adjacent fibers as close as possible, printing speed was relatively low (≈ 4 mm/s), thus limiting the possibility of biofabricating large constructs in an acceptable time.

In order to overcome these constraints, we have developed a custom-made wet-spinning system to generate well-organized myosubstitute with higher resolution (fiber diameter down to approx. 80 μm). Our novel wet-spinning system is notably simpler and easier to use than conventional 3D printers, as it comprises just a co-axial nozzle system and a Teflon drum connected to a stepper motor (Fig 1A–C). The highly aligned skeletal muscle architecture can be achieved by collecting the hydrogel fibers on the rotating drum, without any need of ad hoc printing code. Additionally, the hydrogel fibers that are collected onto the drum form a highly packed bundle with almost no distance between adjacent fibers. This is hardly achievable with conventional extrusion bioprinting approaches (Fig 1D and E).

Another advantageous feature of this system consists of the possibility to fine tune the hydrogel fiber size by controlling the flow rate of the bioink and the rotating speed of the drum. After a thorough characterization of the system (see Fig EV1), we found that hydrogel fibers can be produced in a reproducible manner using a motor speed in the 15 ÷ 70 rpm range (i.e., linear speed ≈ 20 ÷ 92 mm/s), which is 10- to 100-fold faster in comparison with conventional extrusion-based bioprinting approaches (Jian *et al*, 2018), and a bioink flow rate of 50 μl/min. As for the calcium chloride flow rate, we found that values in the range 15 ÷ 25 μl/min were equally suitable for fiber production, with no major impact on fiber size. As anticipated, the diameter of wet-spun fibers and the motor speed are inversely related (Fig 1F). The theoretical values of fiber diameters ($d_\text{fiber}$) could be calculated by assuming a volumetric equivalence with the flow rate of the bioink ($Q_\text{bioink}$) using the formula (Colosi *et al*, 2014):

$$d_\text{fiber} = \sqrt[2]{\frac{4Q_\text{bioink}}{\pi V}}$$

where $V$ is the motor speed. Interestingly, such values were actually higher than the measured ones, for all motor speeds. Such effect was most likely a consequence of alginate shrinking upon exposure to calcium ions and was estimated to be of approximately 30%.

As bioink, we used a formulation already well established by our group containing PEG-fibrinogen (PF), responsible for myogenic differentiation, and alginate (ALG), enabling immediate gelation of spun hydrogel fiber upon exposure to divalent calcium ions. This latter feature was exploited by using a co-axial nozzle extrusion system that allows to supply simultaneously the bioink and the primary cross-linking solution ($CaCl_2$).

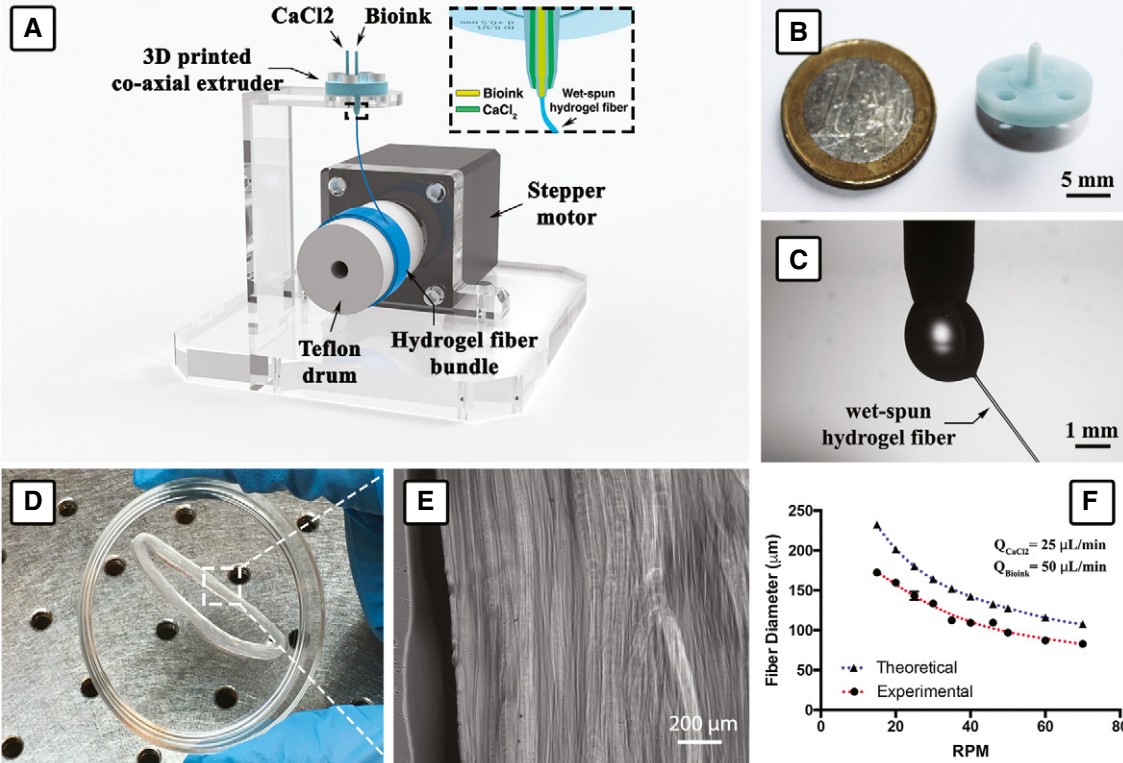

**Figure 1.   Wet-spinning setup.**

A       Schematic representation of the wet-spinning setup that was used for the biofabrication of myo-substitutes.

B       3D-printed co-axial nozzle.

C       Detail of the bioink extrusion at the tip of the co-axial needle.

D, E    (D) Hydrogel fiber bundle after removal from the drum and (E) bright-field high-magnification image showing densely packed fibers.

F       Effect of motor speed on fiber diameter: Interestingly, the experimental values are lower than the theoretical ones due to the shrinking of the alginate-based bioink upon exposure to divalent calcium ions.

## Biofabrication of highly aligned muscle substitutes loaded with murine mesoangioblasts

After system optimization, we exploited the setup for the production of aligned yarns of cell-laden microfibers (Fig 2). In SMTE, next to matrix composition and scaffold architecture, the major role is played by the biological component, i.e., myogenic precursors. In this study, we initially fabricated hydrogel fiber yarns loaded with mouse-derived perivascular progenitor cells, namely mesoangioblasts (Mabs), and tested them for generation of skeletal muscle both *in vitro* and *in vivo*.

Mabs represent a primary myogenic cell population extensively characterized for its capability to robustly differentiate into skeletal muscle *in vitro* and *in vivo* (Díaz-Manera *et al*, 2010; Fuoco *et al*, 2012), which has been also the subject of clinical trial study for duchenne muscular dystrophy (EudraCT Number: 2011-000176-33). As a preliminary test, we encapsulated Mabs within the fibers at different cell densities—ranging from 1 to $5 \times 10^7$ cells/ml. As a result, we found that the best myogenic differentiation was obtained for concentration around $2 \div 2.5 \times 10^7$ cells/ml, a value in agreement with previously published studies (Costantini *et al*, 2017). As shown in Fig 2, Mabs had a typical round morphology upon

encapsulation (day 0); cell elongation and spreading took place since the very first days of culture and were clearly appreciable at day 3 in vast areas of the engineered constructs. This was followed by an abundant myogenesis, with the formation of long-range, multi-nucleated myotubes within two weeks of culture (day 15). These myotubes were characterized by a remarkable aligned architectural organization.

To further validate our approach and especially its cell compatibility toward primary cells, a live/dead staining based on calcein-AM (live cells in green) and propidium iodide (dead cells in red) was performed on the scaffolds at days 1 and 3. In both cases, the majority of Mabs appeared alive, with notable cell elongation at day 3 (see Fig EV2), thus confirming the suitability of our approach for the biofabrication of myo-substitutes.

### *In vitro* characterization of mouse-derived myo-substitutes

In order to assess the effectiveness and advantages of our wet-spinning system in the biofabrication of functional myo-substitute, we compared the *in vitro* myogenesis potential of Mabs in two hydrogel systems: a bulk gel and a wet-spun fiber yarn. The two systems are characterized by the same matrix composition and cell density but

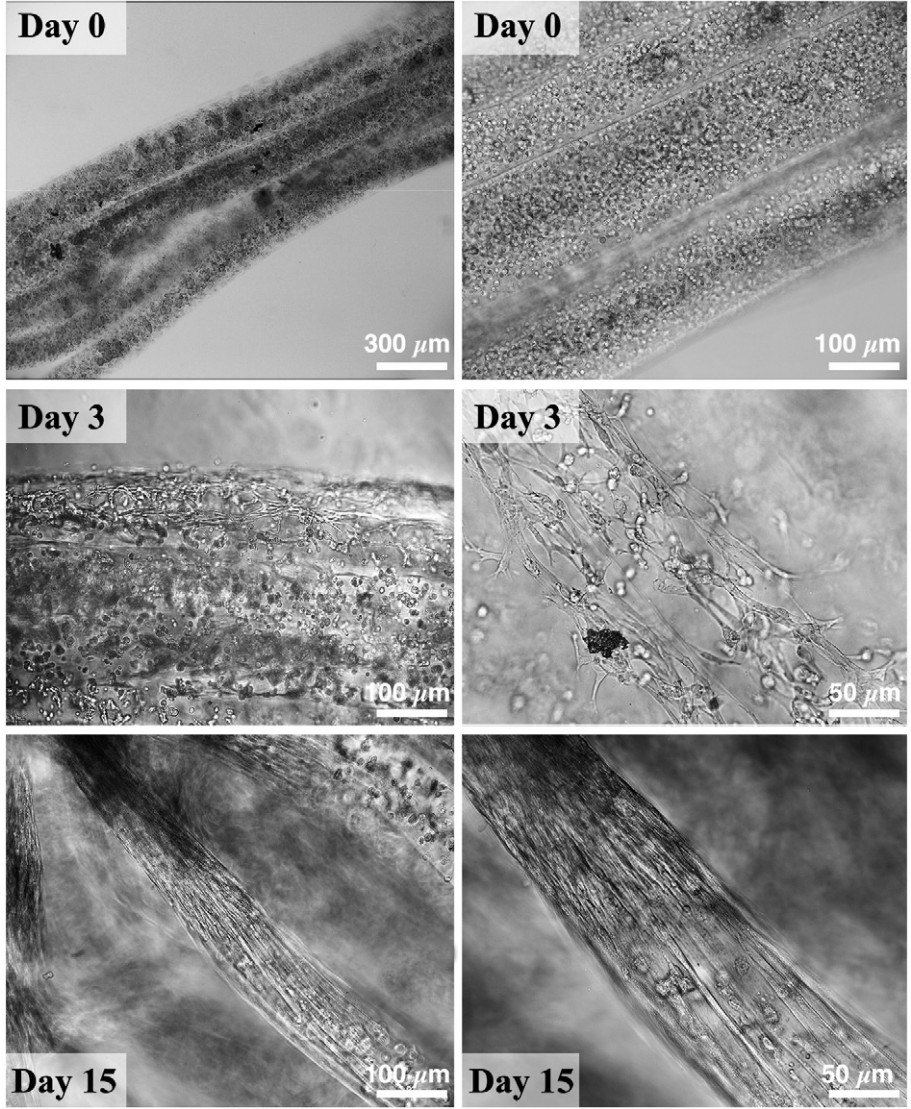

**Figure 2. *In vitro* culture of Mab-loaded hydrogel.**

Mab-laden wet-spun yarns showing progressive myogenic differentiation over culturing time. As can be noted, Mab morphology evolves from a round shape (single cells at early time points) into a more elongated one (generally within the first 3–5 days of culture), eventually forming long-range myotubes. For the sake of clarity, samples used for bright-field investigation were prepared out of few layers of fibers to favor the recognition of cell organization.

significantly different 3D architecture. *In vitro* myogenesis in the engineered myo-substitutes was evaluated by immunofluorescence staining against myosin heavy chain (MHC, a muscle-specific marker highly expressed in fully differentiated myotubes) after 20 days of culture (Fig 3). Mab-laden yarns revealed a substantial myotube parallel organization and positivity for MHC (Fig 3 upper panel), while Mabs embedded in the bulk gels showed a similar MHC expression but an entangled, disordered myotube layout (Fig 3 bottom panel). Being a critical parameter for myo-substitutes functionality, myotube organization was further analyzed quantitatively through fluorescent image analysis. As shown in the polar plots in Fig 3, in Mab-laden yarns all myotubes were distributed within ± 30° of average fiber orientation, whereas in the case of bulk gels, myotube orientation was random with no preferential

directions. Interestingly, we noticed that Mab-laden yarns underwent a partial compaction upon culturing time, reaching approx. a value of 10–15%, a phenomenon often observed in hydrogel laden with skeletal muscle precursors.

## Tibialis anterior (TA) regeneration by means of Mab-derived myo-substitutes

Based on the promising *in vitro* results, Mab-laden myo-substitutes were further tested *in vivo* in a volumetric muscle loss (VML) model following a surgical protocol previously established by our group (Fuoco *et al*, 2015). The approach consists of surgically dislodging around 90% of the tibialis anterior muscle (TA) while leaving the host tendons intact and in place (Fig 4A). In order to precisely

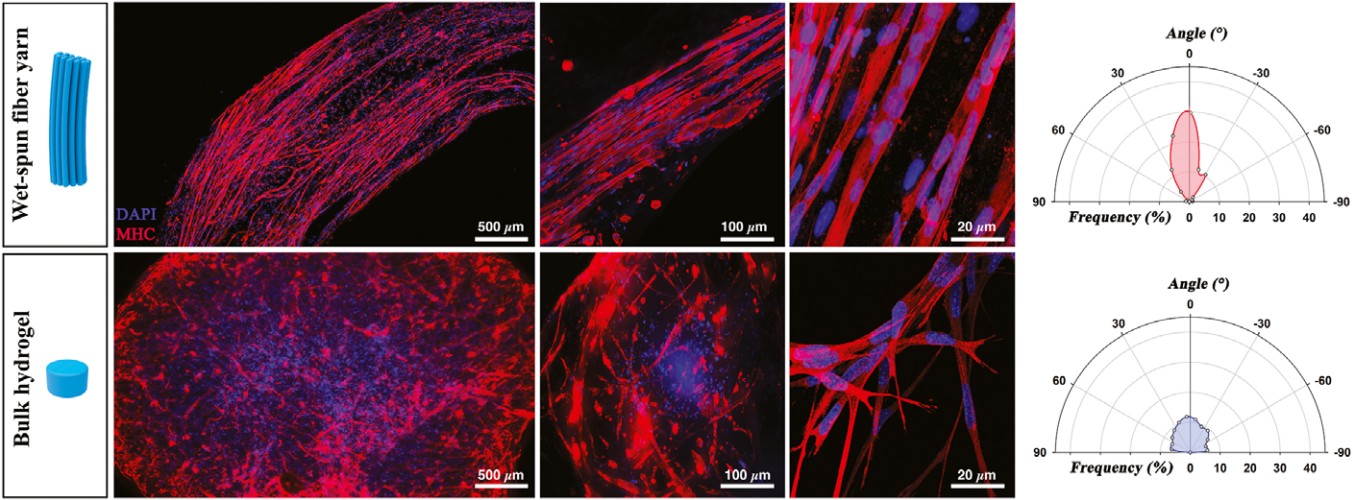

**Figure 3. *In vitro* Mab differentiation.**

MHC staining (red)—performed on 3D bioprinted and bulk structures loaded with Mabs after 15 days of culture—reveals a significant difference in terms of myotube organization between the two hydrogel systems, with the wet-spun constructs greatly outperforming the bulk gels. Such difference—quantified by means of image analysis—is shown in the polar plots. Nuclei were counterstained by DAPI (blue).

distinguish Mab-driven regeneration from that originating from host's cells, the former were labeled by lentiviral transduction with nuclear β-galactosidase (nLacZ) before grafting (Fuoco *et al*, 2012, 2015).

The results in the control animals (untreated) after 20 days post-implantation showed that their leg presented a void where the TA was resected. Most of the missing muscular mass was unable to self-regenerate after VML injury (Fig 4B). Conversely, wet-spun myo-substitute containing nLacZ-expressing Mabs implanted into the contralateral leg yielded a significant muscle regeneration whereby the TA defect was replenished by muscle-like tissue (Fig 4 B). Furthermore, the recovered TA from the treated legs (implanted with wet-spun myo-substitutes) demonstrated larger mass compared with untreated control legs, as evidenced by the measured wet weights of both groups (Fig 4C and D).

Harvested tissues were further analyzed via histological staining to unveil their nature and anatomical structure. Transversal and longitudinal TA sections were firstly processed to detect nLacZ expression employing X-Gal substrate for β-galactosidase activity. Next, samples were immunostained against MHC—for identification of muscle fibers—and laminin (LAM, basal lamina component surrounding muscle fibers), for architectural organization (Fig 5). As already revealed by macroscopic analysis, the cross sections of the control and treated TAs present striking differences in terms of area. Importantly, this difference is predominantly associated with regeneration activity of the implanted Mabs as clearly evidenced by nLacZ staining (Fig 5 upper panel), showing a homogeneous distribution of Mabs throughout the cross section of grafted leg. Moreover, the muscle architectural organization was almost completely reestablished as shown by the longitudinal cross section displaying parallelly oriented MHC-positive myofibers surrounded by laminin (Fig 5 middle and bottom panels). In addition to the reconstituted muscle structure, the magnified views of longitudinal sections underscore the typical hallmarks of sarcomerogenesis and then

functional skeletal muscles (asterisks in Fig 5 bottom panels). Accordingly, the disclosed sarcomeres were detected in close proximity to black X-Gal-labeled nuclei, indicating the nLacZ/Mab-based myo-substitute was able to recapitulate all the morphological features of functional skeletal muscle tissue (arrows in Fig 5 bottom panels). Moreover, despite acellular hydrogel matrix has been already widely tested in our previous study (Fuoco *et al*, 2015), we performed an additional control implanting an acellular wet-spun fiber yarn, which revealed its complete inadequacy in repairing similar VML model (Appendix Fig S1).

Although using immunocompromise mice, we evaluated also the host immune response against the myo-substitute implants. By means of immunofluorescence, we labeled macrophagic infiltration in wt and reconstructed TA at 10 and 20 days after graft (Fig EV3). The obtained results showed a high infiltration rate at day 10, which dropped remarkably at day 20 (Fig EV3G), demonstrating the full integration of the Mab-derived myofibers with the host tissue.

In addition to the muscle-like architectures, there were also uncharacteristic gaps within the muscle tissue, these being noticeable as black voids in the cross sections of the grafted TA. We speculate that these voids are due to the presence of some alginate residues from wet-spun fibers within the engineered muscle, which in physiological conditions degrades slowly and exclusively through the gradual exchange of calcium with monovalent ions (Kong *et al*, 2004). Although this would appear as imperfections in the muscle repair, we hypothesize that such alginate could also exert a beneficial effect in the grafted muscle by acting as pillars, thus rapidly guiding myotube organization around them and providing additional mechanical support to the neo-tissue. Moreover, because of the small fiber size, such gaps do not significantly impair the functionality of the muscle as demonstrated by absolute force measurements of the muscle contraction (see Fig 7).

Regenerated TA derived from nLacZ/Mab-based myo-substitutes showed neo-vascularization—a key feature for artificial tissue

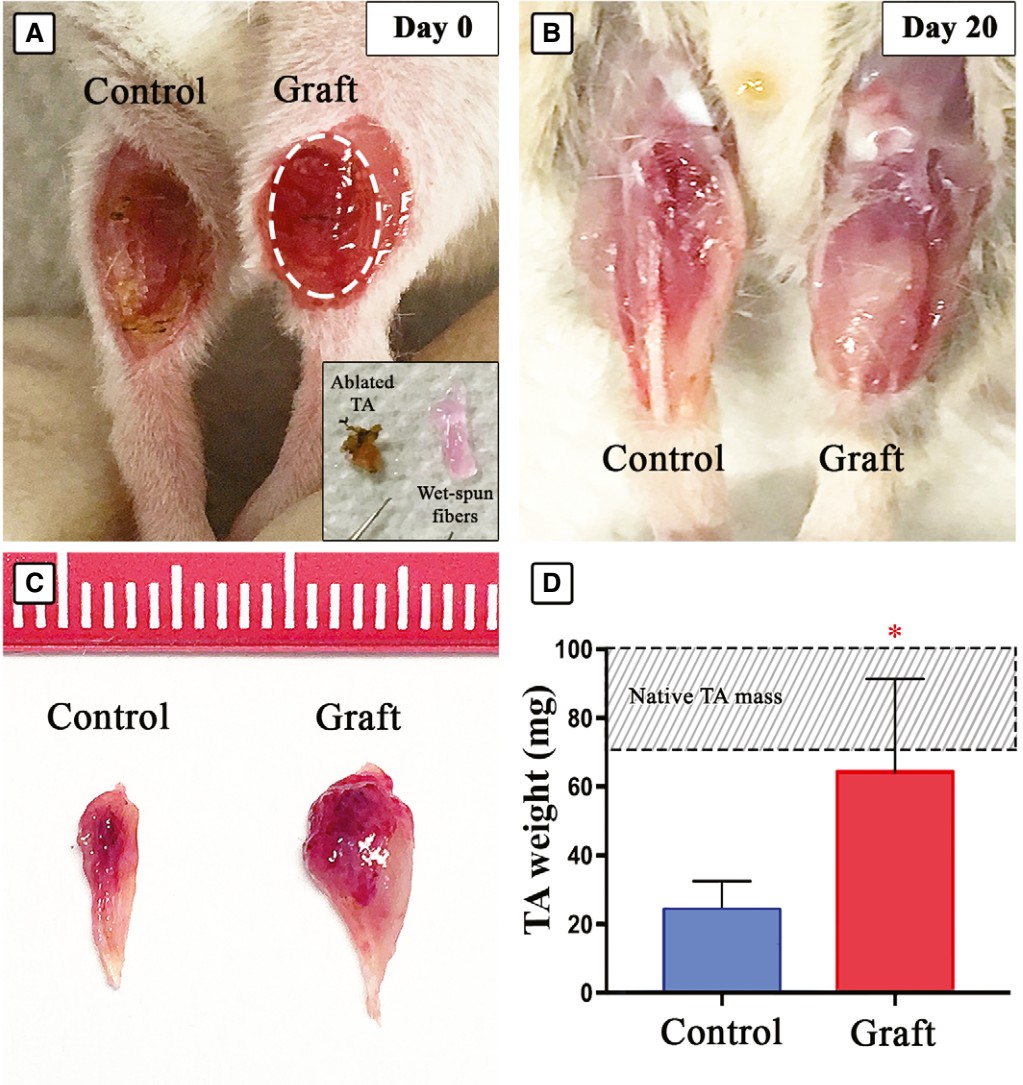

**Figure 4. TA reconstruction using 3D wet-spun myo-substitute.**

A  Immunocompromised mouse leg showing ablated TA (control) and implanted TA (graft) at day 0; inset showing muscle mass removed next to the implanted Mab-loaded constructs shaped to fill the TA lodge.
B  Implanted myo-substitute 20 days after grafting.
C  Isolated TA from control ablated (control) and ablated/grafted leg (graft), the latter revealing a remarkable enhanced size.
D  Graphical representation of the scored TA weight (*n* = 6). The dashed area indicates the range of native TA mass depending primarily on mouse weight and age, and statistical significance was analyzed by ANOVA test (*P* < 0.05 was considered significant: *=0,0306).

subsistence and engraftment (Fig 6). By immunolabeling the explanted grafts for von Willebrand factor (vW, specifically marking endothelial cells) and dystrophin (Dys, a key protein located in proximity of myotube membranes essential for proper muscle functionality, Fig 6A), it was possible to demonstrate that myotubes containing LacZ-positive nuclei were surrounded by small vessel and capillaries (Fig 6B–E). In order to quantify vessel density in the reconstructed TA side, native muscle tissue and reconstructed TA portion (graft) have been compared for vW fluorescent signal area and intensity by ImageJ analysis (see Appendix Fig S2). The evaluation revealed no statistically significant differences among the analyzed muscle moiety, further confirming the good vascularization

of the reconstructed TA (Fig 6D and E). Vessel caliber and numbers were also evaluated using smooth muscle actin (SMA) staining. Despite an apparent larger area of the vessels in the native muscle tissue due to the greater caliber, in the reconstructed TA portion the number of the scored vessels is remarkably higher, confirming vW results and revealing a more than satisfactory vascularization of the reconstructed TA (see Appendix Fig S3).

Furthermore, we have characterized the myofiber composition of the reconstructed TA by means of NADH reductase assay and benchmarked it with native tissue (Dobrowolny *et al*, 2008). The obtained results highlighted no significant differences in muscle fiber composition between native and reconstructed TA after

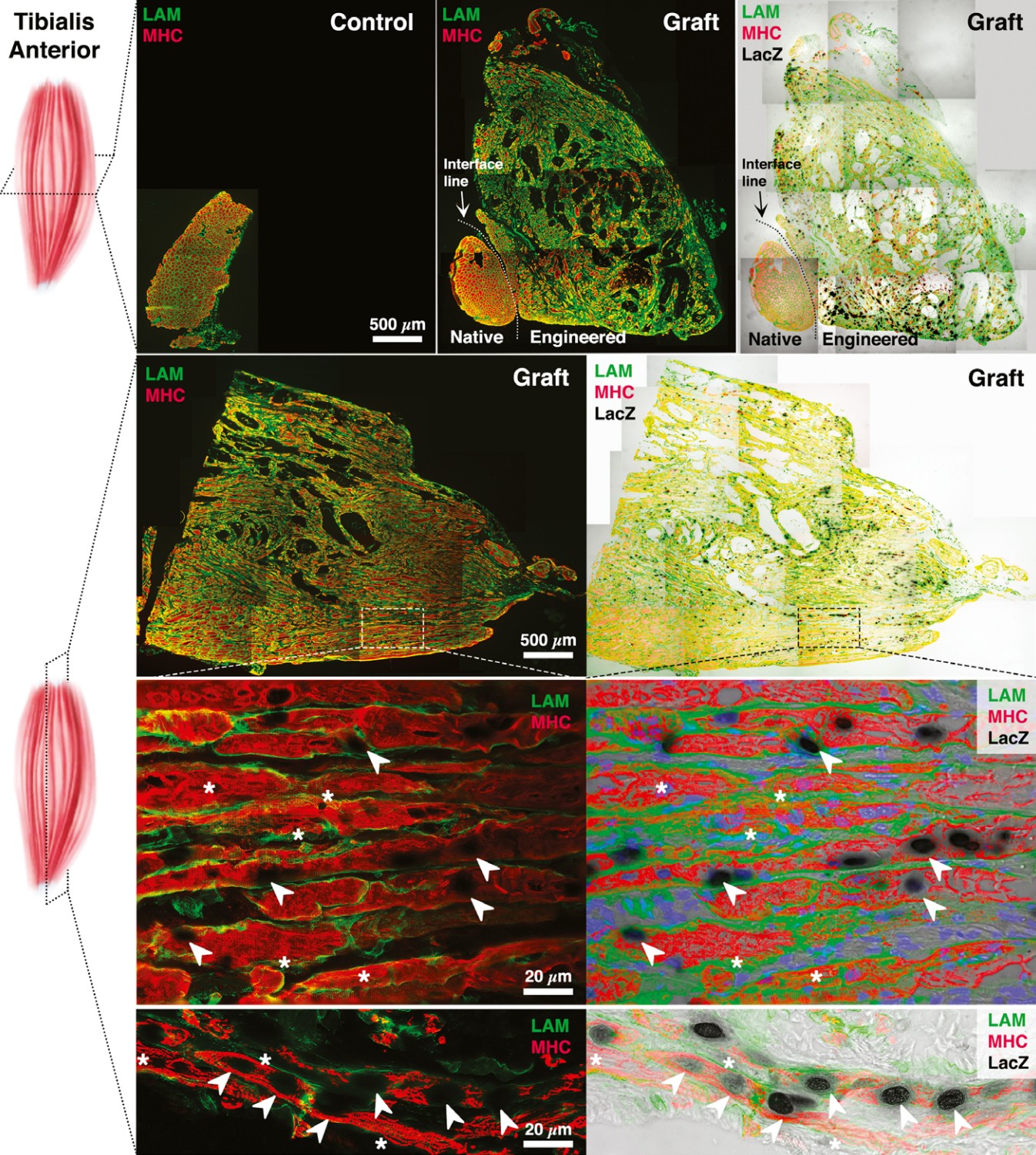

**Figure 5.   Immunofluorescence analysis on ablated TA muscle sections.**

Cross and longitudinal sections of ablated (control) and implanted (graft) TA were marked with LacZ and immunoassayed for MHC (red) and LAM (green). Cross and longitudinal sections were immunostained against MHC and LAM and overlaid to LacZ-labeled images. Dashed area indicates the enlarged view displaying sarcomeres (asterisks) in combination with LacZ-positive nuclei (arrowheads).

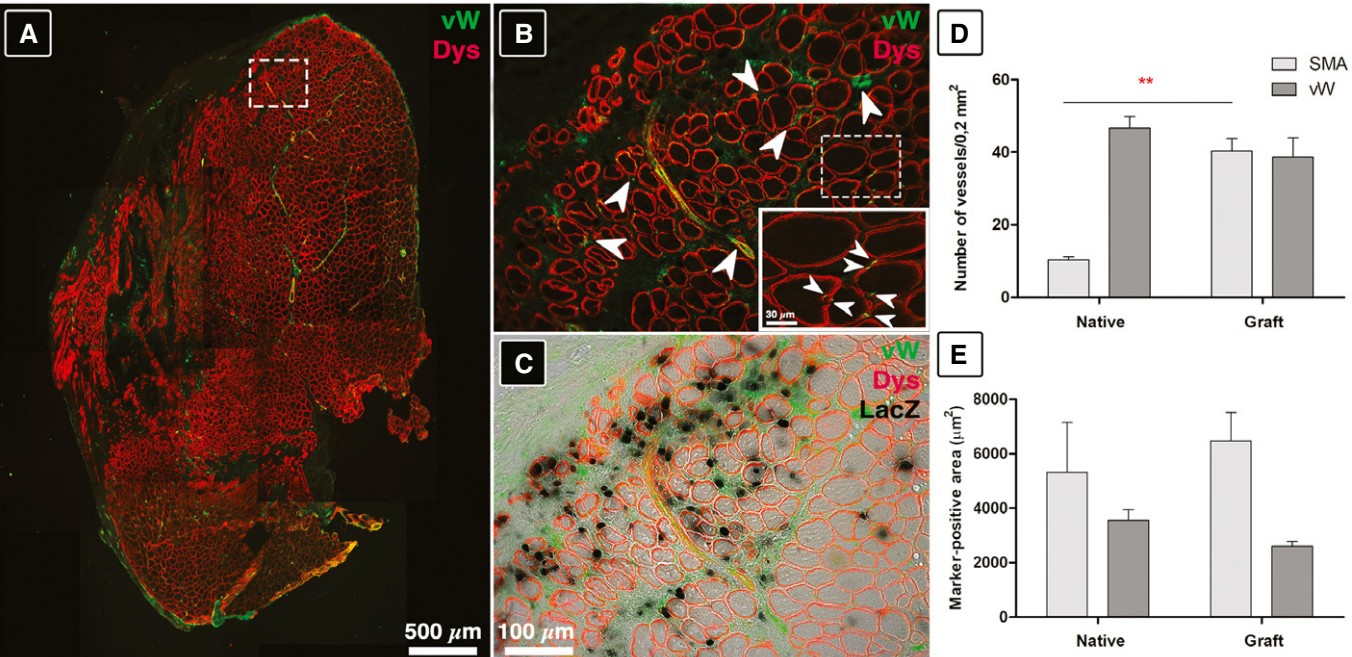

**Figure 6. Neo-vascularization on TA sections implanted with nLacZ/Mab-derived myo-substitutes.**

A    Immunofluorescence characterization of reconstructed TA against vW (green) and Dys (red).
B    Enlarged view of dashed area in (A) white arrows indicate new vessels. Inset: dashed area higher magnification (arrows indicating vWF-positive capillaries).
C    Superimposed image in (B) over LacZ-stained image.
D, E Vessel density assessment by means of fluorescence, and statistical significance was analyzed by ANOVA test ($P < 0.05$ was considered significant: *=0,0019).

20 days from implantation (Fig EV4), showing a very similar rate of oxidative and glycolytic fiber types.

In addition to the morphological analysis, we also carried out a functional assessment of the harvested engineered muscle to evaluate TA muscular activity following massive ablation. Muscle innervation is a necessary condition for ensuring voluntary contraction activation and functionality. Hence, we investigated nerve supply by means of immuno-identification of axons and neuromuscular junctions into engineered muscle fibers (Fig 7). Staining against anti-phospho-neurofilament (NeuF) has been employed to identify motor neuron axons approaching the myofibers. As shown in Fig 7A (marked with an asterisk), NeuF-positive axons inserted between MHC-positive myofibers innervate the engineered TA derived from nLaz/Mabs (Fig 7B). Additionally, simultaneous staining against α-Bungarotoxin (BTX—used to label the acetylcholine receptor in the chemical synapses) and synaptophysin (Syn—labeling neuron major synaptic vesicle protein p38) was used to unveil the formation of numerous functional neuromuscular junctions (NMJs; arrows in Fig 7C). In particular, detailed view of the NMJs from reconstructed TA in the grafted leg reveals the characteristic pretzel-like structure positive for both BTX and Syn (inset in Fig 7C).

To further confirm the functionality of the reconstructed TA, we performed *in vivo* electrophysiological analysis using a well-established experimental setup used for measuring the actual force production (see Appendix Fig S4 and Appendix Table S1; Blaauw *et al*, 2009).

The testing consisted of a direct electrical stimulation of the common peroneal nerve and the simultaneous measurement of the force generated by the TA. The results demonstrated a higher force production in the engineered TA compared with the control, reaching almost a twofold increase in tetanic force for frequencies around 100 Hz (Fig 7D). In agreement to that, approximately 80–90% of the reconstructed TA-NMJs resulted active as shown by the colocalized BTX and Syn immunofluorescence signals (Fig 7E).

Besides all the above analyzed parameters demonstrating the completeness of the reconstructed TA in terms of histoarchitecture and functionality, we have ultimately investigated the replenishment of stem cell niche. Hence, we analyzed by means of immunofluorescence the presence of Pax-7-positive muscle-resident stem cells, namely satellite cells, in the engineered TA muscle. Notably, we identified satellite cells in between LacZ-positive myotubes (arrowheads in Fig EV5), revealing also the provision of myogenic stem cells in the reconstructed mouse TA.

Taken together, the morphological and functional characterization of the engineered TA reveals an unprecedented reconstructive capability of our system, resulting in a swift recovery of the VML defect. The fact that the engineered TA recovered partial functionality only 20 days after implantation further underscores the remarkable speed of the muscle recovery after such a traumatic injury.

### *In vitro* characterization of human-derived myo-substitutes

Mouse-derived myo-structures showed a remarkable capacity in generating functional myo-substitutes *in vitro* and *in vivo*. However, in order to eventually translate such technology to a clinical

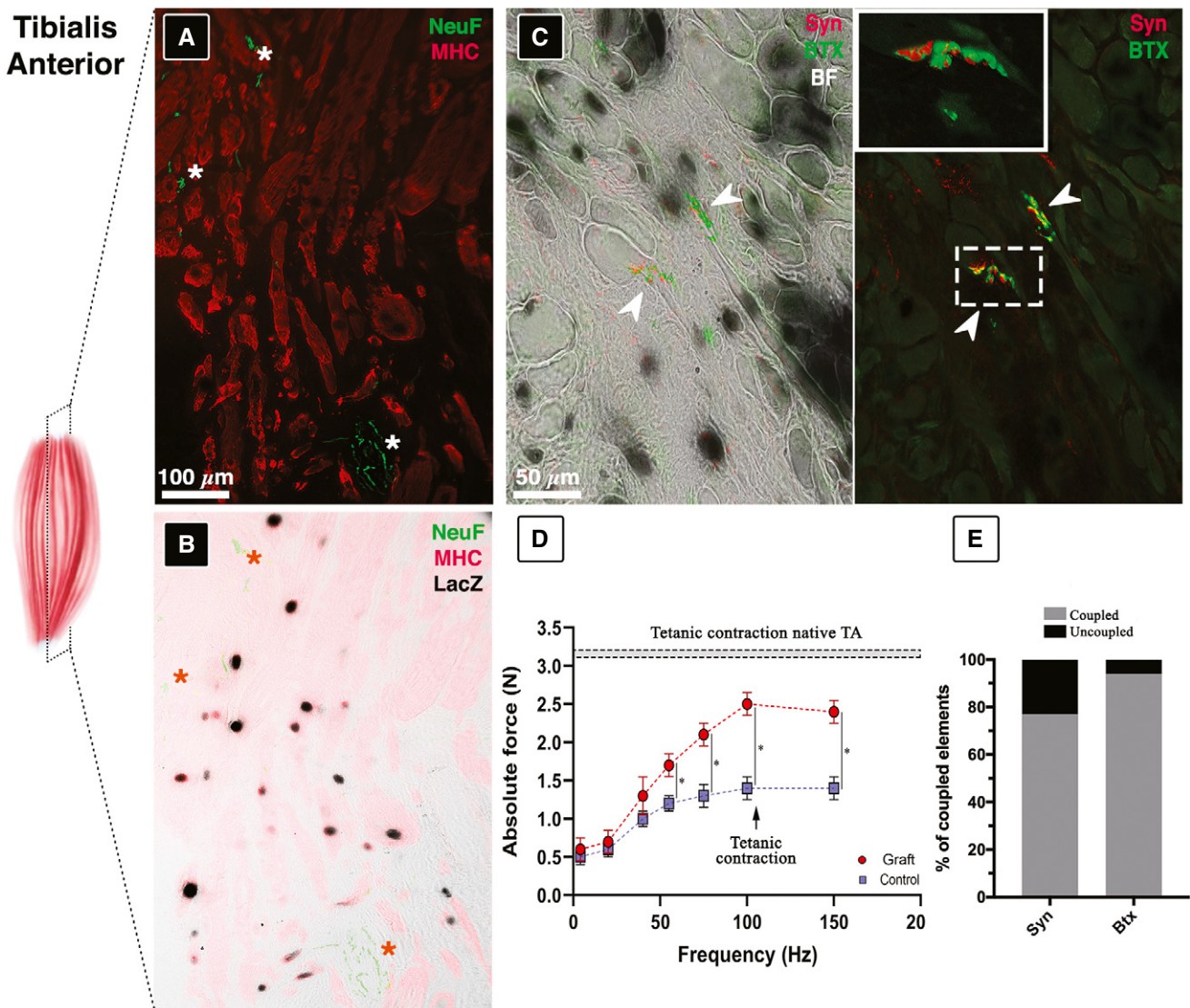

**Figure 7. Innervation assessment on reconstructed TA sections.**

A   Immunofluorescence against NeuF (green) and MHC (red) revealing reconstructed muscle innervation (asterisks).
B   Superimposed image in (A) over LacZ-stained image.
C   Syn and BTX immunolabeled sections showing neuromuscular junction formation (arrowheads), inset enlarged view of dashed box.
D   Electrophysiological analysis scoring muscle strength recovery performed *in vivo* on ablated TA (control) and wet-spun implanted contralateral TA (graft; *n* = 3), and statistical significance was analyzed by ANOVA test ($P < 0.05$ was considered significant: *55 Hz = 0.016830182; *75 Hz = 0.012686264; *100 Hz = 0.009143348; *150 Hz = 0.009327048).
E   Rate of coupled versus uncoupled BTX-Syn signals in reconstructed TA NMJ.

scenario, the performance of the proposed biofabrication approach in combination with human myogenic progenitors needs to be assessed. Stem cell stability and functionality represent a key task for fabrication of large 3D structures (Kang *et al*, 2016). Mechanical forces acting on cells during bioink extrusion—such as high shear stresses—have been demonstrated to directly impact cell proliferation and cell lineage commitment (Bianco & Robey, 2001; Engler *et al*, 2006).

Moreover, different cell lineages, namely mouse Mabs and human primary myoblast, could react differently to the bioink

mechano-chemical properties as extensively demonstrated for other myogenic stem/progenitor cells (Gilbert *et al*, 2010; Nakayama *et al*, 2019). Hence, we performed a preliminary set of *in vitro* experiments with the sole intent of demonstrating the compatibility of the proposed approach with primary human myoblasts (hMyob) and verifying cell stability in terms of myogenic differentiation capacity. To this aim, we benchmarked—as in the case of Mabs—the myo-substitutes fabricated through the proposed wet-spinning technology with standard bulk gel preparations (Fig 8).

The hMyob confined into the tiny fibers underwent myogenic differentiation following a timeline similar to murine Mabs, with a pronounced cell elongation appreciable as early as day 3 (see black arrows in Fig 8, day 3) and an abundant formation of myotube bundles around day 15 of culture (see magnified image of day 15 in Fig 8). The morphological analysis performed 15 days after fabrication by MF20 immunolabeling revealed a substantial myogenic differentiation in both hydrogel systems with an abundant expression of MHC. Also in this case, a significant improvement in the 3D organization of the forming myotubes is noticed in the comparison between the wet-spun fiber yarn and the bulk gel system. As confirmed by the quantitative analysis of myotube orientation shown in the polar plots in Fig 8, the former clearly outperforms the latter.

## Discussion

The *in vitro* fabrication of human tissue equivalents and the *in vivo* restoration of tissue/organ functionalities is one of the biggest biomedical challenges of our times. Thanks to the bio-technological progress of the last two decades, substantial advances in the tissue engineering field have been achieved with the fabrication of liver tissue with sinusoids (Toh *et al*, 2009; Schütte *et al*, 2011), lung functions (Huh *et al*, 2010), spleen (Baker, 2011), skeletal muscle model functionality (Nagamine *et al*, 2011; Serena *et al*, 2016),

*in vitro* models that are capable of mimicking native organ morphology and functionalities.

In the specific context of skeletal muscle tissue engineering (SMTE), researchers have greatly improved myo-substitute 3D morphological organization and functionality by (i) developing specific dynamic culturing protocols, which generally include an electro-mechanical stimulation of the engineered samples, (ii) implementing advanced technologies such as 3D bioprinting or organs on a chip, and (iii) supplying bioactive molecules in combination with tailored matrices to promote myogenic differentiation.

A common limitation shared by all the developed systems consists in the recapitulation of skeletal muscle histoarchitecture and functionality to a limited scale—generally from hundreds of micrometers to a few millimeters. Despite being sufficient for all those applications in which the overall size of the engineered construct does not influence the quality and reliability of the obtained results—such as in the case of miniaturized biohybrid actuators or simplified skeletal muscle models, such construct volumes are far from those needed in a clinical scenario (up to hundreds of $cm^3$). Therefore, the availability of a system enabling the fabrication of macroscopic myo-substitutes that could eventually be translated into the clinics is still an unmet need.

Over the past decade, a condition clinically referred to as volumetric muscle loss (VML) has been the subject of a certain number of studies. A variety of approaches including acellular gels (Quarta *et al*, 2017), cell sheet-derived (Carosio *et al*, 2013), minced muscle

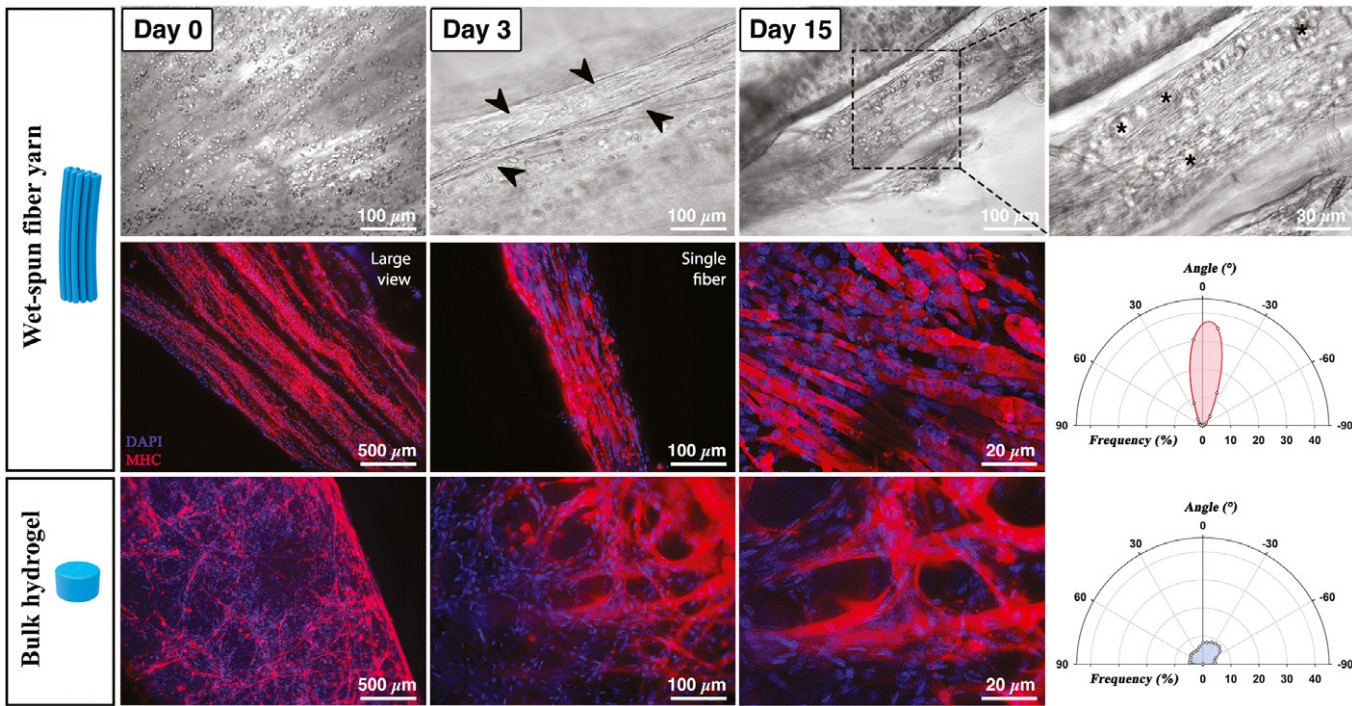

**Figure 8. Fabrication of human myoblast based myo-substitute.**

Wet-spun yarn containing human myogenic precursors showed progressive cell differentiation over time (arrows in day 3); dashed box enlargement reveals still undifferentiated cells after 15 days of culture (upper panel). Middle and lower panel showing MHC immunostaining (red) performed on microfiber yarns and bulk structure loaded with human myoblasts upon 15 days of culture, revealing a significant difference in terms of myotube organization between the two hydrogel systems, with the wet-spun constructs greatly outperforming the bulk gels. Such difference is quantified in the polar plots. Nuclei were counterstained by DAPI (blue).

(Aguilar *et al*, 2018), or tissue engineering grafts (Fuoco *et al*, 2015) have been proposed for VML treatment. Among these strategies, as described in a recent excellent review on this topic (Gilbert-Honick & Grayson, 2019), the use of grafts delivered in combination with a myogenic cell source should be preferred to the other methods. In fact, acellular gels, as ultimately demonstrated in a recent study by Corona (Greising *et al*, 2017), do not offer any beneficial effect for VML recovery. Other strategies suffer from major drawbacks as well, such as scalability, difficulty of handling/surgical implantation (especially in the case of cell sheet), and limited volume of muscle tissue harvestable from the patient. All of these drawbacks complicate the clinical translation for such approaches. On the other hand, tissue-engineered muscle grafts (TEMGs) offer the advantages of enhanced construct durability, ease of surgical manipulation, and high scalability potential. However, due to the lack of a standard

reference VML model, comparison between studies is not trivial due to the broad range of myogenic precursors and scaffold materials as well as differences in *in vitro* pre-culturing time, construct size, and type of VML model—i.e., animal used, size of VML defect, muscle anatomy, and pennation. Therefore, in order to compare more effectively our findings with the state of the art, we have selected only those studies in which a VML defect was created in the tibialis anterior of small-sized animals—i.e., mice and rats. A brief overview of the results described in these studies is reported in Table 1.

As can be observed in the table, a different VML model was employed in each study with the damage size ranging generally between 20% and 40% of TA volume (with the exception of the study from Fuoco *et al* in which the VML defect size reached approx. 90% of the TA volume); implant duration then ranged from 3 weeks to 6 months. Such variability of VML

**Table 1. State of the art of TA-VML reconstruction in small size animals (mice and rats)**

| Animal model | Damage size | Myogenic cells | Matrix composition | Implant type | Implant duration | **Characterization** | | | Ref. |
|---|---|---|---|---|---|---|---|---|---|
| | | | | | | Vascularization | Innervation | Force recovery | |
| | | | | | | *Explants positive for:* | | | |
| Mouse TA | 80–90% | Mabs | PEGylated fibrinogen | Gel casted *in situ* | 6 months | SMA, VE-cad | NeuF, α-BTX | Comparable with WT (native muscle). | Fuoco *et al*, 2015 |
| Rat TA | 20% | Acellular | Urinary bladder matrix (UBM) | Graft | 4 months | vWF | n.a. | No significant difference compared with control (injured but not treated muscle). | Aurora *et al*, 2015 |
| Mouse TA | 25% | MDSCs | Fibrin | Gel-casted *in situ* | 1 month | vWF | n.a. | n.a. | Matthias *et al*, 2018 |
| Rat TA | 30% | C2C12 or hMPCs | Fibrin | Gel injected *in situ* multiple times | 1 month | vWF, SMA | NeuF, AChR | Treated legs recovered 40% of force of native TA muscle. | Kim *et al*, 2016 |
| Rat TA | 30% | Rat MPCs | Cell sheets supported by bone-tendon anchors | Graft | 1 month | CD31 | NeuF, α-BTX | Engineered TA generated 75% of force with respect to control (native muscle) | VanDusen *et al*, 2014 |
| Mouse TA | 20% | C2C12 or hMPCs | Collagen type I nanofibrillar strips | Graft | 3 weeks | CD31 | n.a. | n.a. | Nakayama *et al*, 2019 |
| Mouse TA | 40% | Human or murine MuSCs | Muscle-derived dECM | Graft | 1 month | CD31 | α-BTX | Engineered TA generated 60% of force with respect to control (native muscle) | Quarta *et al*, 2017 |
| Rat TA | 20% | Autologous minced muscle graft | Collagen | Graft | 2 months | n.a. | n.a. | Engineered TA generated 48% of force with respect to control (native muscle) | Aguilar *et al*, 2018 |

models—compounded by the use of different scaffold materials and myogenic precursors—makes it difficult to rank the different approaches. Nevertheless, there are a few outcomes that stand out. For example, the worst results in terms of distribution of force recovery were obtained for acellular implants (no significant difference compared with control leg), while the most promising results were achieved with a PEGylated fibrinogen gel loaded with Mabs casted directly *in situ* (force comparable with WT native muscle) 6 months after implantation.

Despite the promising outcomes in force and muscle mass recovery, the obtained results are still unsatisfactory due to the recurring problems related to cell engraftment, differentiation efficacy, muscle fibrosis, long regeneration time (up to 6 months), and limited system scalability (Fuoco *et al*, 2016a,b; Quarta *et al*, 2017). Scalability—being a pivotal requirement for translating animal models into human clinical therapy—is a critical feature that has so far often limited the possibility to translate the aforementioned approaches into VML models of large-sized animals (Greising *et al*, 2017).

In this context, our work proposes an easily scalable SMTE approach, which is completely independent from muscle defect geometry, based on a simple yet robust biofabrication wet-spinning strategy that enables the assembly of large myo-substitutes capable of efficiently guiding and speeding up cell differentiation and 3D muscle tissue organization both *in vitro* and *in vivo*. These features are successfully achieved by assembling highly anisotropic bundles composed of tiny hydrogel fibers where myogenic precursors are geometrically confined. In spite of their simplicity, such architectures cannot be fabricated using other fiber-based techniques—e.g., 3D bioprinting—due to technical limitations such as printing speed and resolution (Costantini *et al*, 2017).

In our previous work, we have reported an advanced reconstruction of a functional mouse TA (Fuoco *et al*, 2015); nevertheless, a long implantation time (6 months) was needed to reestablish TA mass, morphology, and functionality. Moreover, the proposed strategy relied on the polymerization *in situ* of a liquid hydrogel-based precursor solution—a strategy suitable for the TA lodge thanks to its concave shape but hardly transferable to other muscle locations.

Here, we have demonstrated that the aforementioned limits can be overcome by employing a biofabricated wet-spun fiber yarn loaded with primary muscle precursors. Such constructs, in fact, can be easily manipulated, thus being totally independent from the transplantation muscle shape and size. Moreover, these constructs have shown an extraordinary capacity in regenerating muscle mass, vasculature, and innervation, thus allowing recovery of muscle functions within only 20 days after implantation upon a massive TA ablation (~90% muscle mass removed). Last, but definitely not least, the preliminary evidences of compatibility of our approach with human primary myoblast open an actual possibility to translate this approach into a clinical scenario.

Currently, we are investing a great deal of work to improve construct matrix remodeling *in vivo* by precisely tuning bioink composition and to incorporate vasculature and innervation features during the biofabrication process. The latter aspects, in fact, are undoubtedly needed to successfully translate our approach into large animal VML models. In addition, we are planning to test in the near future our approach for VML recovery in a swine model using autologous cells.

In conclusion, in light of the presented results, we think that the proposed strategy represents a breakthrough in the field and, with further refinements, may be successfully translated to the clinic as a reconstructive therapy for VML patients.

# Materials and Methods

### Materials

All chemicals were purchased from Sigma-Aldrich and used without further purification unless otherwise stated. Sodium alginate (ALG, Mw 33 kDa) was a kind gift from FMC Biopolymers. Photocurable PEG-fibrinogen was synthesized following previously published protocols (Almany & Seliktar, 2005).

### Design and fabrication of wet-spinning setup

Cell-laden hydrogel fibers were fabricated using a custom wet-spinning approach (Rinoldi *et al*, 2019). The whole system is composed of a co-axial nozzle 3D printed via stereolithography (DWS, DIGITALWAX 028J, Italy, using a THERMA DM210 photoresin) and a stepper motor control through an Arduino UNO electronic board. The characteristic dimensions of the co-axial nozzle were as follows: ID 0.3 mm and OD 1.3 mm. The internal needle protrudes out from the external one of approximately 500 μm (Colosi *et al*, 2014). After 3D printing, the nozzle was thoroughly rinsed in ethanol to remove unreacted photoresin and then dried at room temperature overnight. Prior to use, the co-axial nozzle was additionally coated with parylene (coating thickness ≈ 2 μm) using a chemical vapor deposition machine (Parylene Deposition System 2010, Specialty Coating Systems Inc.). Finally, the 3D-printed needle was assembled on top of a polycarbonate structure and placed centrally with respect to the rotating Teflon drum connected to the stepper motor shaft.

### Biofabrication of 3D constructs

#### Bioink formulation
All the experiments were performed using an alginate-based bioink (ALG = 4% w/v) blended with photocurable PEG-fibrinogen (PEG-Fib = 0.8% w/v). Biopolymers were dissolved in a 25 mM HEPES buffer solution, and 0.1% w/v Irgacure 2959 was added to the bioink as radical photoinitiator. Skeletal muscle progenitors (human primary myoblasts or murine mesoangioblasts) were resuspended in the bioink to a final concentration of $2 \times 10^7$ cells/ml. Such cell density has been thoroughly optimized in our previous work (Costantini *et al*, 2017).

#### Bulk hydrogel fabrication
3D bulk hydrogels were fabricated by a casting method. Briefly, bioink solution loaded with cells was poured into cylindrical silicon molds (PDMS) and then exposed first to a $CaCl_2$ solution (to promote physical alginate cross-linking) and then for 5 min to UV light (to polymerize PEG-fibrinogen).

#### 3D wet-spun fiber construct fabrication
Wet-spun hydrogel fiber scaffolds were fabricated using the custom setup described in paragraph 2.2. Prior to performing experiments with cells, the wet-spinning system was deeply characterized to

investigate the relations between motor speed, fluid flow rates (bioink and calcium chloride solution), and fiber size (see Fig EV1). A typical experiment consists of supplying the bioink through the inner nozzle and a calcium chloride solution (0.3 M) through the external one. When the two solutions come into contact at the tip of the co-axial nozzle system, an instantaneous gelation of the bioink solution occurs. The resulting hydrogel blob is then pulled gently with a tweezer—forming a hydrogel fiber—until it reaches the surface of the rotating Teflon drum (diameter = 25 mm) connected to the motor shaft. As soon as the fiber touches the drum, a hydrogel fiber starts to be continuously extruded from the nozzle and collected onto the drum forming a bundle. After careful optimization, we decided to fabricate hydrogel fibers of around 100 μm. Fluid flow rates and motor speed were adjusted as follows: bioink flow rate = 50 μl/min, calcium chloride flow rate = 25 μl/min, and motor speed = 50 rpm (linear speed $\approx$ 65 mm/s). Samples were collected for approximately 2 min forming constructs of an overall volume of approx. 60 μl, which contained $1.2 \times 10^6$ cells/sample. After that, they were UV cross-linked to stabilize the PEG-fibrinogen and cultured for 3 days *in vitro* before implantation. During the *in vitro* pre-culturing time, samples were kept as rings. At the moment of implantation, samples were sterile-cut with a scalpel to fill the lodge of the mouse TA, which was previously removed.

### *In vitro* cell and cell-laden sample culture

nLacZ-modified mouse mesoangioblasts (Mabs) were obtained as already described in previous work (Díaz-Manera *et al*, 2010; Fuoco *et al*, 2012). Briefly, Mabs were transduced by nLacZ lentiviral particles and cultured on Falcon dishes at 37°C with 5% $CO_2$ in DMEM GlutaMAX (Gibco) supplemented with heat-inactivated 10% fetal bovine serum (FBS), 100 international units/ml penicillin, and 100 mg/ml streptomycin. Human muscle-derived primary myoblasts were isolated and cultured as previously described (Vianello *et al*, 2017). Briefly, muscle biopsies from healthy donors were minced and digested in 0.8% w/v collagenase I (Life Technologies, Carlsbad, CA, USA) in DMEM, supplemented with 100 U/ml penicillin, and 100 mg/ml streptomycin (Life Technologies, Carlsbad, CA, USA), for 60 min. Afterward, muscle fragments were gently dissociated by pipetting and passing through a 21 G syringe needle. Cell suspension was centrifuged for 10' at 300 *g*, and the pellet was resuspended and seeded on a Matrigel-coated 35-mm Petri dishes in growth medium composed of 20% FBS, 25 ng/μl hFGFb (human basic fibroblast growth factor, ImmunoTools; Friesoythe, Germany) in Ham's F12 medium (EuroClone; Milan, Italy) with Pen/Strep. Cells were expanded and cultured in 100-mm Petri dishes. During *in vitro* culture, cell-laden constructs were cultured exclusively in DMEM GlutaMAX (Gibco) supplemented with heat-inactivated 10% fetal bovine serum (FBS), 100 international units/ml penicillin, and 100 mg/ml streptomycin.

### *In vivo* construct implantation

Two-month-old male SCID/Beige mice were anesthetized with an intramuscular injection of physiologic saline (10 ml/kg) containing ketamine (5 mg/ml) and xylazine (1 mg/ml) and then the 3D constructs were implanted in tibialis anterior muscle (TA), according to following surgical procedure (Fuoco *et al*, 2015): (i) Limited incision on the medial side of the leg has been performed in order to reach the TA, (ii) utilizing a cautery to avoid bleeding, the muscle fibers were deeply removed (leaving in place the tendons) to create a venue for the implant, and (iii) the pre-shaped construct was placed in the removed TA fibers lodge and the incision was sutured. As control, the contralateral TA was surgically ablated, but no construct was implanted. Analgesic treatment (Rimadyl, Pfizer, USA) was administered after the surgery to reduce pain and discomfort. Mice were sacrificed 20 days after implantation for molecular and morphological analysis. Experiments on animals were conducted according to the rules of good animal experimentation I.A.C.U.C. No 432 of March 12, 2006, and under Italian Health Ministry Approval No. 228/2015-PR.

### Force measurements

Contractile performance of treated and control muscles was measured *in vivo* using a 305B muscle lever system (Aurora Scientific, Aurora, ON, Canada) in animals anesthetized with a mixture of tiletamine, zolazepam, and xylazine. Mice were placed on a thermostatically controlled table, with knee kept stationary and foot firmly fixed to a footplate, which, in turn, was connected to the shaft of the motor. Contraction was elicited by electrical stimulation of the peroneal nerve. Teflon-covered stainless-steel electrodes were implanted near the branch of the peroneal nerve as it emerges distally from the popliteal fossa. The two thin electrodes were sewn on both sides of the nerve, and the skin above was sutured. The electrodes were connected to an AMP Master-8 Stimulator (AMP Instruments, Jerusalem, Israel). Isometric contractions induced at different frequencies were performed to determine the force–frequency relationship.

### Histology and immunostaining

*In vitro* 3D constructs after 20 days of culture were fixed in PFA 2%, while *in vivo* grafts were processed for histology as previously described (Costantini *et al*, 2017). Briefly, 3D constructs were surgically explanted, embedded in O.C.T., and quickly frozen in liquid nitrogen-cooled isopentane for sectioning at a thickness of 10 μm on a Leica cryostat. Then, *in vitro* and *in vivo* resulting samples were processed for immunofluorescence. As primary antibodies, we used anti-myosin heavy chain (mouse monoclonal, DHSB, diluted 1:2), anti-laminin (rabbit polyclonal; Sigma-Aldrich, # L9393, diluted 1:100), anti-Dystrophin Rod Domain (mouse monoclonal; Vector Laboratories, # VP-D508, diluted 1:100), anti-von Willebrand factor (rabbit polyclonal; Dako, # A0082, diluted 1:100), anti-Pax7 (mouse monoclonal, DHSB, diluted 1:4), anti-Neurofilament H phospho (rabbit polyclonal; BioLegend, # PRB-573C, diluted 1:400), anti-Mannose Receptor (rabbit polyclonal; Abcam, # ab64693), and anti-Synaptophysin (mouse monoclonal; Abcam, # ab8049). As secondary antibodies, we used anti-mouse Alexa Fluor 555® (goat polyclonal; Invitrogen, # A32727, diluted 1.200) and anti-rabbit Alexa Fluor 488® (goat polyclonal; Invitrogen, # A32723, diluted 1:200). LacZ-positive cells were detected with beta-galactosidase staining (Abcam, # ab102534), while acetylcholine receptors were detected by α-bungarotoxin (Alexa Fluor 488®; Invitrogen, # B13422). Nuclei were stained with 300 nM DAPI (Thermo Fisher Scientific, # D1306). Images were acquired using a Nikon Eclipse TE-2000 fluorescent microscope equipped with UV source and CoolSNAP Photometrics CCD Camera and a Nikon A1R + Laser Scanning Confocal Microscope.

## Live/Dead assay

Cell viability in 3D wet-spun constructs was assessed 1 and 3 days after fabrication, using a cell stain Double Staining Kit (Sigma-Aldrich), which allows the simultaneous fluorescence staining of viable and death cells. Briefly, after incubation of constructs with calcein-AM (viable cells) and propidium iodide (dead cells) solutions for 30 min at 37°C, live and dead cells were displayed and photographed with Nikon Eclipse 2000-TE Fluorescence Microscope.

## Image analysis

### Alignment evaluation

The alignment of the differentiated muscle fibers was assessed using OrientationJ plugin for ImageJ/FIJI (Püspöki et al, 2016). The local orientation and isotropic properties, such as coherency, are evaluated for every pixel of the image based on structure tensors (Rezakhaniha et al, 2012). In order to analyze the images, we converted them in highly contrasted 8-bit images and processed them using OrientationJ dominant direction and OrientationJ distribution tools. OrientationJ dominant direction computes the coherency value, an index ranging between 0 and 1, with 1 indicating highly oriented structures and 0 indicating isotropic areas. OrientationJ distribution processes a weighted histogram of orientations, where the weight is the coherency itself.

### Vessel density evaluation

To obtain a quantification of the vascularization levels in the native muscle tissue and in the engrafted one, images from immunofluorescence analysis against SMA (smooth muscle actin) and vW (von Willebrand factor), well-known vascular markers, were analyzed using ImageJ program. The average fluorescence intensity of each immunofluorescence image was calculated as mean gray value (*Analyze > Set Measurement > Mean Grey Value*, then *Analyze > Measure*), an index elaborated by the program by assigning to each pixel a grayscale value. The sum of the gray values of all the pixels in the image is then divided by the total number of pixels. As a result, the higher the mean gray value, the higher the average intensity of the fluorescent signal in the image. The total SMA-positive area was calculated on thresholded images (*Image > Adjust > Threshold*) using the command *Analyze Particles* and represents the total number of pixels that are part of the selection, spatially calibrated in squared μm (*Analyze > Set Scale*). The results are indicative for the SMA expression extent in the tissue. The percentage of the SMA-positive area was calculated on thresholded images (*Image > Adjust>Threshold*) using the command *Analyze Particles* and represents the percentage of pixels that are part of the selection, indicative for the SMA expression extent in the tissue. Lastly, the number of SMA- and vW-positive vessels was manually counted in selected ROIs of 0.2 mm$^2$.

### Macrophage quantification

The macrophage percentage quantification is based on a modified version of the Di Vito and colleagues method (Di Vito et al, 2015), using ImageJ program. Briefly, a 200 × 200 μm area was selected in TA images in three different conditions (wt, 10 days, and 20 days after graft implantation), to calculate macrophage percentage as the ratio between the area including mannose receptor-positive cells and the total area of the ROI.

### The paper explained

#### Problem

Skeletal muscle regenerative capability is related to damage size, in other words small injuries are able to self-repair while larger lesions need therapeutic treatment and/or surgical intervention. Muscular dystrophies and large trauma due to both accidental tearing and cancer ablation surgery lead to extensive loss of tissue, requiring a reconstructive approach to recover the muscle damage. Within this scenario, skeletal muscle tissue engineering represents a real opportunity to replace damaged or lost muscular tissue with complete and functional implantable artificial muscle. But in spite of everything, there is still an open challenge related to the architectural muscular organization replica and overall the functionality in terms of vessel and nerve supply.

#### Results

Exploiting homemade 3D printing technology to assemble in orderly way myogenic precursor cells and supporting matrix, we were able to replace large damaged muscle mass and rapidly restore (within 3 weeks) muscular histoarchitecture and functionality in a mouse model of extensive muscle damage. Forcing parallel alignment by 3D printing innovative approach, macroscopic artificial muscular structures were generated in vitro. Upon implantation, the printed muscle substitutes supported the formation of new blood vessels and neuro-muscular junctions, fundamental for artificial muscle survival and functional recovery.

#### Impact

These data represent a solid base for further testing the 3D printing technology here proposed in large animal size, before to eventually be translated into clinical scenarios for the treatment of a wide range of pathology eliciting muscle wasting.

### Pre- and post-synaptic element colocalization

The evaluation of the pre- and post-synaptic element colocalization was obtained by manually counting all the synaptophysin-positive structures, all the bungarotoxin-positive structures, and the number of each marker-positive structures colocalizing with the other markers. These values were used to calculate the rate of synaptophysin-positive structures colocalizing with bungarotoxin and *vice versa*.

### Muscle fiber composition

NADH reductase assay was performed on transversal of 10 μm TA cryosections. Briefly, after rehydrating, sections were immersed for 30 min at 37°C on a solution of Tris buffer 0.05 M pH 7.5 containing NADH disodium trihydrate salt (Life Science) and 4-nitro blue tetrazolium chloride (NBT; Sigma-Aldrich). The sections were rinsed with ddH$_2$O and then dehydrated with increasing concentrations of acetone (Analyticals, Carlo Erba). The acetone was removed, and the samples were mounted to be counted. The counting was performed manually considering the relative number of oxidative fibers versus the total, and no statistical differences were observed.

## Statistical analysis

The animal sample size was chosen to guarantee the precision of our estimates and the study power for correct result interpretation in line with the 3R guidelines. Any inclusion/exclusion criteria have been employed for the study, and all the samples or animals were analyzed in blind when possible. All experiments were performed in biological

triplicate. Data were analyzed using GraphPad Prism 7, and values were expressed as means ± standard error (SEM). Statistical significance was tested using one-way ANOVA and Student's *t*-tests. A probability of less than 5% ($P < 0.05$) was considered to be statistically significant.

## Data availability

This study includes no data deposited in external repositories.

**Expanded View** for this article is available online.

## Acknowledgements

We would like to thank Giulio Cossu for critical reading. This study was supported by the National Centre for Research and Development under the frame of BIOMOTION project (PL-TW/VI/3/2019) to WŚ, the National Science Centre Poland (NCN) within SONATA 14 Project No. 2018/31/D/ST8/03647 to MC, the Italian Ministry of University and Research PRIN Funding Scheme No. 2015FBNB5Y_002 to CG and 201742SBXA_004 to SC, and the Italian Ministry of Defense by the Funding Scheme No. PNRM 2019 for the project RESUMO No. 2018.019 to CG.

## Author contributions

MC and CG conceived and designed the experiments; MC, ST, EF, and CF carried out experiments; MC, CG, LC, SC, and GC supervised the project. AR acquired fluorescence images relative to *in vitro* culture; MC, MN, WS, PG, and ST conceived, designed, fabricated, and optimized the co-axial wet-spinning platform; SB and CSR performed histology and NADH-TR essay, and LV isolated and cultured human primary myoblast beside critical reading; JB, CZ, and RB provided surgical support for TA muscle ablation; BB performed electrophysiological analysis together with critical reading; and DS provided PEG-fibrinogen characterization and production next to critical reading. MC, ST, EF, and CG wrote the manuscript. All co-authors contributed to discussion and analysis of the data.

## Conflict of interest

The authors declare that they have no conflict of interest.

The authors agree to make available to colleagues in academic research all new reagents, including organisms (or means to produce them), viruses, cells, nucleic acids, and antibodies that were used in the research reported and that are not available from public repositories or commercial suppliers. Human patient samples and data will be made available in accordance with the relevant ethical standards.

## For more information

https://www.mtec-sc.org/volumetric-muscle-loss-vml-repair-following-extre mity-trauma/

https://clinicaltrials.gov/ct2/show/NCT04051242

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
