## [Review Process File · EMBO Molecular Medicine]

Biofabricating murine and human myo-substitutes for rapid volumetric muscle loss restoration

Marco Costantini, Stefano Testa, Ersilia Fornetti, Claudia Fuoco, Carles Sanchez Riera, Minghao Nie, Sergio Bernardini, Alberto Rainer, Jacopo Baldi, Carmine Zoccali, Roberto Biagini, Luisa Castagnoli, Libero Vitiello, Bert Blaauw, Dror Seliktar, Wojciech Świąszkowski, Piotr Garstecki, Shoji Takeuchi, Gianni Cesareni, Stefano Cannata, and Cesare Gargioli

DOI: [10.15252/emmm.202012778](https://doi.org/10.15252/emmm.202012778)

Corresponding author: Cesare Gargioli (cesare.gargioli@uniroma2.it)

Review Timeline:

Submission Date:	19th May 20
Editorial Decision:	30th Jun 20
Revision Received:	9th Nov 20
Editorial Decision:	22nd Dec 20
Revision Received:	5th Jan 21
Accepted:	12th Jan 21

Editor: Zeljko Durdevic

Transaction Report:

Dear Dr. Gargioli,

Thank you for the submission of your manuscript to EMBO Molecular Medicine. We have now received feedback from the three reviewers who agreed to evaluate your manuscript. As you will see from the reports below, the referees acknowledge the interest of the study but also raise partially overlapping concerns that should be addressed in a major revision. Particular attention should be given to the control experiments using biomaterials only (without cells) and to the degree of innervation and macrophage infiltration.

Addressing the reviewers' concerns in full will be necessary for further considering the manuscript in our journal, and acceptance of the manuscript will entail a second round of review. EMBO Molecular Medicine encourages a single round of revision only and therefore, acceptance or rejection of the manuscript will depend on the completeness of your responses included in the next, final version of the manuscript. For this reason, and to save you from any frustrations in the end, I would strongly advise against returning an incomplete revision.

We realize that the current situation is exceptional on the account of the COVID-19/SARS-CoV-2 pandemic. Therefore, please let us know if you need more than three months to revise the manuscript.

I look forward to receiving your revised manuscript.

***** Reviewer's comments *****

Referee #1 (Remarks for Author):

The authors of the paper by Costantini et al extended previous work improving the technical approach to obtain biofabricated muscle-like structure.

The study is well performed and the paper is clearly written. The data may have relevance to the development of treatment strategies for the muscle disease, considering the evidence that this structure displays an extraordinary capacity in regenerating muscle mass.

Nevertheless, some point remains to be elucidated and additional results are needed to strength

the conclusions.

1) The authors reported, by means of immunofluorescence, the presence of Pax-7 positive muscle resident stem cells, namely satellite cells, in the engineered TA muscle.

Is this an endogenous population of stem cells or alternatively the myofabricated structure maintains some undifferentiated Pax7 positive cells? How do the authors discriminate the origin of this stem cell population? What is the percentage of Pax7 positive cells within the graft? Are these Pax 7 positive cells also present in the myofabricated structure before the in vivo transplantation? Does this population actively participate to muscle regeneration? In this context would be interesting to verify the presence of embryo-MyHC positive fibers.

2) The authors reported that In vitro myogenesis in the engineered myo-substitutes was evaluated by immunofluorescence staining against myosin heavy chain. Which isoform of MyHC is expressed (fast, slow, both)? Is there any difference in the expression of MyHC isoforms between in vitro cultures and after myofabricated structure transplantation? Does the innervation change/modulate the expression of MyHC isoforms?

3) The authors reported that the myofabricated structure shows typical signs of sarcomerogenesis. It is quite difficult to appreciate, based on the performed analysis, the presence of sarcomeric organization within the myostructure. The electron microscopy or alternatively confocal microscopy analysis, using relevant markers of the sarcomeres, would strength this conclusion.

4) The authors reported the formation of numerous neuromuscular junctions (NMJ) in the grafted myostructure. This is one of the most important and intriguing part of the study, suggesting a functional integration of the grafted myostructure and the establishment of the nerve-muscle interplay. Nevertheless, it is quite difficult to detect real innervated fibers, considering that there is significant proportion of NeuF/BTX positive elements that do not colocalize. It is worth to show the percentage of innervation. In this context, the authors can incubate the myosections with α -bungarotoxin and with synaptophysin and neurofilament antibodies to define the percentage of innervated, intermediate and denervated endplates as the total number of endplates divided by the number of innervated (overlapping synaptophysin and α -bungarotoxin staining), intermediate (no overlap but synaptophysin staining adjacent to α -bungarotoxin) or denervated (no overlap and no apparent synaptophysin staining adjacent to α -bungarotoxin staining) endplates. Moreover, the authors can verify the ratio between the gamma and epsilon subunits expression of AChR, considering that AChR-gamma expression increases in denervated muscle or under conditions that alter the NMJ functionality, whereas AChR-epsilon is expressed in mature innervated fibers.

Minor point

Supplementary Figure 3 should be Figure 5

Referee #2 (Remarks for Author):

The authors describe how a wet-spun myo-substitute graft seeded with myogenic progenitor can drive efficient muscle regeneration after an extreme muscle laceration and ablation in mice. This protocol can in principle be applied to human since the myo-substitute hydrogel fibers can support the growth and differentiation of human myogenic progenitor as well.

There are some experimental details that I found missing. What is the dimension of the myo-substitute, and how many MABs are initially seeded and grown and for how long before the grafting? Were they differentiated before the grafting or grafting was performed with proliferating cells?

The control of the graft was only the ablated muscle, that was unable to regenerate itself. Since in the grafted animal we can observe many lacZ negative myotubes, it is conceivable that endogenous myogenic precursors are recruited to the regeneration site. Would the myosubstitute hydrogels fibers without any cells grown or with non-myogenic cells be sufficient to allow a certain amount of colonization and regeneration by the host cells? What is the proportion of muscle of host vs seeded Mabs origin?

Graft have been implanted in immunodeficient SCID beige mice, that should nevertheless have quite normal macrophages. Have macrophage infiltration in the graft been analyzed at earlier time points?

The myo-substitute graft is able to be vascularized and innervated and some electrophysiological assays demonstrated a certain degree of functional recovery of the muscle after 20 days. Possible other functional test on muscle motor function could be performed also at later time points.

Referee #3 (Remarks for Author):

The paper by Costantini addresses the need for an implantable muscle therapy. Their approach is interesting as they use a wet spinning setup for in vitro muscle construction, the product of which they term myo-substitute. The wet spinning setup is a quick way to produce aligned fibers, with a dramatic increase in extrusion speed compared to the bioprinting approach described in Costantini et al 2017. Murine as well as human myogenic cell containing myo-substitutes are fabricated and the murine ones are implanted in three mice which suffered volumetric muscle loss. Almost full function recovery after 20 days is described. The manuscript raises some methodological questions as well as concerns of data interpretation.

The authors already described a very similar study: the construction of a PEG-fibrinogen plug containing mesoangioblasts with cross-linking in situ (Fuoco et al 2015). In the current discussion, the wet spinning approach is deemed better than the in situ cross-linking approach for two reasons. First, the 2015 approach was deemed too time-consuming: "a long implantation time (6 months) was needed to reestablish TA mass". However, although indeed the 2015 study had a follow up of 6 months, a functional recovery was apparent by 2 weeks (Fuoco et al 2015 figure 4 grip test results compare black to red datapoints). Second, the 2015 strategy is described as "a strategy suitable for the TA lodge thanks to its concave shape but hardly transferable to other muscle locations". It is not clear why the strategy would not be transferable to other muscle locations. This rationale deserves more explanation. As authors also acknowledge in the discussion, the strategy as currently presented does not yet overcome the main problems the field is facing for creation of larger pieces: a functional vasculature and innervation. Therefore, it is a bit of a stretch to state that "the proposed strategy represents a breakthrough in the field".

A control with biomaterials only -without cells- would be needed to assess the effect of the engineered muscle rather than autologous repair due to host cell migration. The rationale not to perform this is based on previous work (Fuoco et al 2015). It is stated that the acellular matrix is

completely inefficacious, however from the referenced work, it appears that the matrix alone has some benefit over control, although less than with cells added (Fuoco et al 2015 figure 4 grip test green data points). In the current approach, not only the matrix composition is different (addition of alginate), but also the fiber extrusion creates a different scaffold architecture. Therefore, it would be interesting to include such control. Even when less effective than with cells included, the use of a biomaterial only would be of high interest and would be more straightforward to translate to a clinical therapy. Supporting an important contribution of the host cells, the majority of transplanted LacZ positive cells seem to reside outside of the muscle (dystrophin) areas (Figure 6c). The same can be observed in Figure 5, where most nuclei are LacZ negative and also many of those are in MHC+ areas. And the same can be observed in Fig 7 (compare panel a and b): quite a number of MHC positive cells do not contain a lacZ nucleus. Additionally, from figure S5 (wrongly referenced as supplementary figure 3 in the manuscript) the Pax7+ satellite cells in the myo-substitute are not LacZ+, indicating that also these stem cells derive from the host.

Authors describe that the wet-spinning system was deeply characterized to investigate the relations between motor speed, fluid flowrates (bioink and calcium chloride solution) and fiber size. Can the data relating fluid flowrates to fiber size be shown ? (perhaps as supplementary information)

"After careful optimization, we decided to fabricate hydrogel fibers of around 100 μm ." Likely, authors mean 100 μm . Which parameter was optimized ?

The size of the myo-substitute right after collection from the drum is not specified. What is the diameter of the drum ? From the drum, a ring structure is obtained. Such ring structure presumably is not what was implanted. How was the ring-shaped myo-substitute further treated ? Was there any in vitro culture of the myo-substitute before implantation ? Was the whole ring structure implanted or a part of it ? If the latter, how much ? How does the ring-shape relate to the structure of the myo-substitute shown in Fig 4a (inset) ?

The myo-substitutes are described to contain aligned myotubes in culture after 2 weeks (P14). It would be good to guide the reader to fig 3 to support the claim. What were the culture conditions in which this was carried out ? Was any further shrinkage (after the initial 30%) of the myo-substitutes observed in vitro ? The human myo-substitutes seem to shrink dramatically between day 0 (Fig 8, second row, left) about 1 mm and day 3 (fig 8 second row second column) about 100 μm . Is this correct or is this some imaging artefact ?

From fig 2 it appear that the thickness of the myo-substitute is approximately 300-600 μm at day 0. This seems very thin as an implant. Also, the extruded ring is collected for 2 minutes at 50 rpm, which means 100 fibers of 100 μm thickness each; this would presumably be thicker ? In fig 2, only about 5 parallel fibers can be discerned. And in Fig S1, the myo-substitute seems even more thin, since 1 cell layer is relatively well in focus without much signal from cells above or below.

The graft seems to expand in size quite considerably in vivo. From Fig 3 left image (in vitro) the size of the myo-substitute seems to be in the range of 1 mm. In Fig 4c, the thickness after 20 days in vivo seems to be around 4 mm. How can such increase be explained ? Is size of the graft perhaps in part due to oedema and/or macrophage cell infiltration ? In Fuoco et al. 2015, authors described that "even in an immunodeficient background, over time the xenogeneic graft with time attracted murine macrophages and other non-lymphoid cells that infiltrated and prevented myogenic maturation of the new muscle". From what mouse species were mouse MABs derived ? Authors make reference to Fuoco et al 2012, but there the species was also not described. In the latter

paper, authors reference Minasi et al 2002. These MABs were derived from the dorsal aorta. Moreover, the animals used in Minasi were not SCID animals. Would it be possible that a xenogeneic response was mounted against the MABs ?

At what doubling were cells (human and mouse) used ? Can some basic characterization of the human cells be shown (eg % myogenic cells) ?

For the surgical procedure, was the TA completely ablated except for about 5% remaining on both ends attached to the tendon ? Or is the remaining 10 % still connecting the tendons ? It is unclear how the grafted myo-substitute is attaching to the remaining tendon. Especially given the force generated; 2N is considerable and it is intriguing how such force could be generated if not properly attached to the tendons. Characterization of the myotendinous junction would be of significant benefit to this study.

It is impossible to discern a sarcomere structure in the areas designated with the asterisks in Fig 5.

A widespread laminin staining can be observed after implantation (Fig 5). This is very encouraging and may hint towards a very active extracellular matrix remodeling, almost completely replacing the PEG-fibrinogen. However, it would be needed to include the necessary controls to exclude cross-reactivity/ background staining of the laminin antibody or autofluorescence of the used biomaterials.

In the legend of Fig 5, authors state that myo-substitutes showed neo-vascularization, however, no markers of vascularization are presented in this figure.

Fig 6: what is the inset in panel b ?

The methods section contains several sentences on the SMA quantification. However, it is not clear how the quantification of pixels translates to the quantification of number of vessels as presented in Figure 6. The mere presence of SMA+ cells does not prove the presence of functional blood vessels. From supplementary figures 2 and 3, SMA positive cells can be seen, however a lumen can in most cases not be discerned. Moreover, the SMA may be derived from MABs differentiated to smooth muscle cells. Supporting such interpretation is the co-localisation of vWF and LacZ staining on Fig 6c.

Why is there a large discrepancy between SMA and vWF staining in native muscle for number of vessels (Fig 6d) ? In contrast, in Fig 6e, it appears that more surface is SMA positive than vWF positive.

Fig 6c: As pointed out before, very few of the dystrophin+ areas (myofibers) seem to contain lacZ (transplanted cell) nuclei. However, (interestingly !) a number of green areas (vWF, endothelial cells) seem to be lacZ positive. This may indicate that part of the mesoangioblasts have differentiated to endothelial cells. This would warrant further investigation and if true, may provide a positive rationale to the use of mesoangioblasts.

Fig6d is a quantification of the number of vessels per 0.2 mm². This area corresponds more or less to the image area shown in Fig 6b (0.5x0.4 mm) as judging from the scalebar in panel a). It seems a bit of a stretch to identify 40 vessels in this figure. It rather seems that only 1 vessel-like structure (with an identifiable lumen) can be observed.

In Fig 7b, it seems that the areas where neuF is observed are relatively far apart from a lacZ

nucleus. Therefore, wouldn't this be indicative of host-derived myofibers which are in proximity to axons (NeuF staining) ?

In the functional test (Fig 7d), the data for a native muscle are shown as a grey zone. It is unclear what this zone is based on. To compare data and as a positive control, it would be needed to show actual measurements for the native situation with the used experimental setup.

Fig 8: please label panels. In the figure of the bulk hydrogel, second to the left, it seems that many nuclei are MHC negative. Does this mean that the used cells are to a large extent non-myogenic ?

Fig S1 : please provide scale bars.

Word errors:

P1 The sentence starting with "Of note ..."is grammatically not correct

P2: a reliable...models -> model

Consistently use either mesoangioblasts or mesangioblasts

P6: isolated and culture -> isolated and cultured

P11: $P < 0.05$ -> $P < 0.05$

P11: unprecedent -> unprecedented

P12 easier to use then -> easier to use than

P15: we have not leaved out -> we have not left out

Point by point letter**Referee #1 (Remarks for Author):**

1) The authors reported, by means of immunofluorescence, the presence of Pax-7 positive muscle resident stem cells, namely satellite cells, in the engineered TA muscle. Is this an endogenous population of stem cells or alternatively the myofabricated structure maintains some undifferentiated Pax7 positive cells? How do the authors discriminate the origin of this stem cell population? What is the percentage of Pax7 positive cells within the graft? Are these Pax 7 positive cells also present in the myofabricated structure before the in vivo transplantation?

Authors' reply: We thank the Reviewer for his/her comment. Mabs do not express Pax7 *in vitro*. In our previous paper (Fuoco et al., 2015), we have discriminated host satellite cells from Mabs-derived satellite cells by colocalization in immunofluorescence for LacZ and Pax7 demonstrating that a small part of implanted Mabs are able to replenish satellite cells pool. Moreover, labeling implanted cells by X-Gal staining, it makes impossible the simultaneous staining with pax7 antibody because of the precipitate in the nucleus, furthermore, to the best of our knowledge, there are no more reliable antibodies against LacZ commercially available. Thus it is impossible to address this issue. Nevertheless, we believe that this information could provide a marginal detail because, independently of the origin (host or graft), the most important outcome of our study is that satellite cells are present in the engineered tissue so as to guarantee muscle tissue homeostasis. Of note, within implanted acellular construct (Appendix Fig. S1) we did not observe any tissue self-regeneration neither pax7 positive cells.

Does this population actively participate to muscle regeneration? In this context would be interesting to verify the presence of embryo-MyHC positive fibers.

Authors' reply: As above stated, pax7 positive cells populate reconstructed TA only when Mabs are encapsulated in the myo-substitute. This event can be explained due to the direct differentiation of Mabs into satellite cells, as previously demonstrated in other studies (Fuoco et al., 2015 and Dellavalle et al., 2007) or the migration from remaining and/or surrounding native muscle (Hughes and Blau, 1990). We believe that the positive staining for mature MyHC obtained 20 days after engraftment makes unneeded the one for embryo-MyHC.

2) The authors reported that In vitro myogenesis in the engineered myo-substitutes was evaluated by immunofluorescence staining against myosin heavy chain. Which isoform of MyHC is expressed (fast, slow, both)? Is there any difference in the expression of MyHC isoforms between in vitro cultures and after myofabricated structure transplantation? Does the innervation change/modulate the expression of MyHC isoforms?

Authors' reply: Following Reviewer criticism, we performed NADH reductase assay on wild type TA and reconstructed TA upon myo-substitute engraftment in order to evaluate the muscle fiber type composition (Figure EV4). NADH-TR staining revealed a very similar composition of fiber types, in terms of oxidative slow type I fiber rate versus fast glycolytic type II fiber. We skip *in vitro* MyHC isoform analysis because we believe that *in vitro* myo-structure and reconstructed TA upon implantation are not comparable due the high degree of maturation *in vivo*, for this reason we performed NADH reductase assay between wt and reconstructed TA.

3) The authors reported that the myofabricated structure shows typical signs of sarcomerogenesis. It is quite difficult to appreciate, based on the performed analysis, the presence of sarcomeric organization within the myostructure. The electron microscopy or alternatively confocal microscopy analysis, using relevant markers of the sarcomeres, would strength this conclusion.

Authors' reply: Accordingly, we have modified Figure 5 inserting a more enlarged view of a LacZ positive myofibers showing clear sign of sarcomeres represented by perpendicular streaks labeled by sarcomeric heavy myosin (MF20 DHSB antibody).

4) The authors reported the formation of numerous neuromuscular junctions (NMJ) in the grafted myostructure. This is one of the most important and intriguing part of the study, suggesting a functional integration of the grafted myostructure and the establishment of the nerve-muscle interplay. Nevertheless, it is quite difficult to detect real innervated fibers, considering that there is significant proportion of NeuF/BTX positive elements that do not colocalize. It is worth to show the percentage of innervation. In this context, the authors can incubate the myosections with α -bungarotoxin and with synaptophysin and neurofilament antibodies to define the percentage of innervated, intermediate and denervated endplates as the total number of endplates divided by the number of innervated (overlapping synaptophysin and α -bungarotoxin staining), intermediate (no overlap but synaptophysin staining adjacent to α -bungarotoxin) or denervated (no overlap and no apparent synaptophysin staining adjacent to α -bungarotoxin staining) endplates.

Authors' reply: We thank the Reviewer for the comment. Accordingly, we have performed synaptophysin/BTX staining in order to evaluate the rate of innervate endplates. The data relative to this analysis have been added in Figure 7. As one can notice, syn/BTX staining and the relative colocalization quantification demonstrated a quite full innervation of the endplates.

Moreover, the authors can verify the ratio between the gamma and epsilon subunits expression of AChR, considering that AChR-gamma expression increases in denervated muscle or under conditions that alter the NMJ functionality, whereas AChR-epsilon is expressed in mature innervated fibers.

Authors' reply: We thank the Reviewer for the comment, nevertheless this type of characterization related to AChR subunits is exclusively performed in *in vitro* NMJ modelling to assess the maturation degree, which, in our case, has been widely demonstrated with the syn/BTX staining and *in vivo* electrophysiology.

Minor point

Supplementary Figure 3 should be Figure 5

Authors' reply: We thank the Reviewer for his/her criticism; we have modified the figure numbers.

Referee #2 (Remarks for Author):

There are some experimental details that I found missing. What is the dimension of the myo-substitute, and how many Mabs are initially seeded and grown and for how long before the grafting? Were they differentiated before the grafting or grafting was performed with proliferating cells?

Authors' reply: According to Reviewer's comment, we have introduced the missing information about wet-spun sample preparation in the paragraph 2.3 *Biofabrication of 3D constructs*. The overall volume of a sample can be calculated from the data displayed in the paragraph using the following formula:

$$V_{sample} = r_{fiber}^2 \cdot \pi \cdot s \cdot t_{collection} \quad \text{eq.1}$$

Where s is the linear speed of the drum and $t_{collection}$ is the collection time of the sample. The linear speed can be calculated from the rotating drum speed using this formula:

$$s = \frac{\omega_{rpm} \cdot 2\pi \cdot r_{drum}}{60} = \frac{50 \cdot 2\pi \cdot 12.5mm}{60s} \approx 65 \text{ mm/s}$$

Using this value in eq. 1, one can calculate the volume of a sample

$$V_{sample} = r^2 \cdot \pi \cdot s \cdot t_{collection} = (0.05mm)^2 \cdot 3,14 \cdot 65mm/s \cdot 120s = 49 \text{ mm}^3 = 61 \text{ mL}$$

Therefore, keeping in mind that we fabricated samples using a cell density of 2×10^7 cells/mL, each sample contained approximately 1.2 million cells ($n. \text{ cells}_{sample} = V_{sample} \times \text{cell density}$).

After fabrication, samples were pre-cultured for 3 days *in vitro* and then implanted in TA. The rationale behind this decision is that within such pre-culturing time cells can start elongating while they can complete the differentiation process *in vivo*. Nevertheless, we expect part of the cells to be still in a proliferating state at the moment of implantation.

The control of the graft was only the ablated muscle, that was unable to regenerate itself. Since in the grafted animal we can observe many lacZ negative myotubes, it is conceivable that endogenous myogenic precursors are recruited to the regeneration site. Would the myosubstitute hydrogels fibers without any cells grown or with non-myogenic cells be sufficient to allow a certain amount of colonization and regeneration by the host cells? What is the proportion of muscle of host vs seeded Mabs origin?

Authors' reply: Following Reviewer's 2 and Reviewer's 3 criticism, we have performed acellular construct implantation to assess the effect of the hydrogel matrix itself in recruiting host cells. In Appendix Fig. S1 we analyzed acellular PEG-Fibrinogen-based construct implantation after 10 and 20 days upon graft. The experiment performed demonstrated that without cellular component there is not regeneration at all neither endogenous cell recruitment. The void matrix

is almost completely resorbed within 10 days from implantation, revealing only a small leftover, marked by the dashed rectangle in Appendix Fig. S1a.

Graft have been implanted in immunodeficient SCID beige mice, that should nevertheless have quite normal macrophages. Have macrophage infiltration in the graft been analyzed at earlier time points?

Authors' reply: In agreement with the Reviewer, we performed macrophage immunolabeling employing Mannose Receptor antibody (Anti-Mannose Receptor antibody (ab64693) Abcam). In Figure EV3 we quantified macrophagic behavior in wt and reconstructed TA after 10 and 20 days from myo-substitute implantation, revealing their initial high infiltration rate at day 10 which dropped remarkably at 20 day.

Possible other functional test on muscle motor function could be performed also at later time points.

Authors' reply: We thank the Reviewer for his/her consideration. Indeed, further extending the present study with additional tests performed possibly also at later time point is of great interest. However, we believe that the presented results are already satisfactory and robust to be published without any other characterization.

Referee #3 (Remarks for Author):

The authors already described a very similar study: the construction of a PEG-fibrinogen plug containing mesoangioblasts with cross-linking *in situ* (Fuoco et al 2015). In the current discussion, the wet spinning approach is deemed better than the *in situ* cross-linking approach for two reasons. First, the 2015 approach was deemed too time-consuming: "a long implantation time (6 months) was needed to reestablish TA mass". However, although indeed the 2015 study had a follow up of 6 months, a functional recovery was apparent by 2 weeks (Fuoco et al 2015 figure 4 grip test results compare black to red datapoints). Second, the 2015 strategy is described as "a strategy suitable for the TA lodge thanks to its concave shape but hardly transferable to other muscle locations". It is not clear why the strategy would not be transferable to other muscle locations. This rationale deserves more explanation. As authors also acknowledge in the discussion, the strategy as currently presented does not yet overcome the main problems the field is facing for creation of larger pieces: a functional vasculature and innervation. Therefore, it is a bit of a stretch to state that "the proposed strategy represents a breakthrough in the field".

Authors' reply: We thank the Reviewer for this observation. We believe that using precursor solution for *in situ* casting have limited applicability because of the intrinsic limits related with the shape and position of the damaged muscle. Moreover, the use of an isotropic bulk gel often leads to an unwanted, entangled myotube organization, similar to that presented in the manuscript. Using 3D printing technology represent a win-win situation as, from the one hand, enables to process the same hydrogel matrix and, from the other, introduces a much higher level of control in terms of 3D cell organization in the matrix and achievable matrix architectures which can mimic precisely the histoarchitecture of the native skeletal muscle. Of note, the geometrical confinement itself drives cell organization into parallelly organized myotubes accelerating significantly the myo-substitutes maturation.

It is true that in Fuoco et al., 2015, the grip test revealed a partial strength recovery after 2 weeks, but, from the histological point of view, the muscle was only partially formed.

In the present study, by exploiting an innovative and straightforward 3D printing method, we have obtained a complete, vascularized and innervated TA muscle within 20 days from myo-substitute implantation. Moreover, 3D printing approach does not present limits in terms of size and shape to be printed enabling additionally the precise spatial control of cell distribution within the 3D myo-substitutes. For these reasons, we truly believe that the statement "the proposed strategy represents a breakthrough in the field" is not an overestimation of the proposed approach. Thanks to the promising presented results, we are now exploiting this method to reconstruct skeletal muscles in swine models.

A control with biomaterials only -without cells- would be needed to asses the effect of the engineered muscle rather than autologous repair due to host cell migration. The rationale not to perform this is based on previous work (Fuoco et al 2015). It is stated that the acellular matrix is

completely inefficacious, however from the referenced work, it appears that the matrix alone has some benefit over control, although less than with cells added (Fuoco et al 2015 figure 4 grip test green data points). In the current approach, not only the matrix composition is different (addition of alginate), but also the fiber extrusion creates a different scaffold architecture. Therefore, it would be interesting to include such control. Even when less effective than with cells included, the use of a biomaterial only would be of high interest and would be more straightforward to translate to a clinical therapy. Supporting an important contribution of the host cells, the majority of transplanted LacZ positive cells seem to reside outside of the muscle (dystrophin) areas (Figure 6c). The same can be observed in Figure 5, where most nuclei are LacZ negative and also many of those are in MHC+ areas. And the same can be observed in Fig 7 (compare panel a and b): quite a number of MHC positive cells do not contain a lacZ nucleus. Additionally, from figure S5 (wrongly referenced as supplementary figure 3 in the manuscript) the Pax7+ satellite cells in the myo-substitute are not LacZ+, indicating that also these stem cells derive from the host.

Authors' reply: We agree with the Reviewer criticism and accordingly we have performed acellular construct implantation. In Appendix Fig. S1 we analyzed acellular PEG-Fibrinogen-based construct implantation after 10 and 20 days upon graft. The experiment performed demonstrated that without cellular component there is not regeneration at all neither endogenous cell recruitment. The void matrix is almost completely resorbed within 10 days from implantation, revealing only a small leftover, marked by the dashed rectangle in Appendix Fig. S1a.

Hence, as stated in 2015, acellular construct unfortunately does not promote any cell recruitment neither muscle regeneration. Regarding LacZ positive nuclei, we added few new images for the supplementary experiment required (Fig.5 lower panel high magnified myofibers, new Fig.7, Fig. EV3) showing high LacZ positive nuclei integration in the muscle fibers. Moreover, in the new lower panel in figure 5, one can notice that several LacZ+ nuclei are quite faint, explaining why in some sections it seems there are just few Lacz+ cells. Regarding Figure 6, we agree that there are several Lacz+ nuclei outside the fibers, but we would like to stress the point that Mabs are perivascular progenitor cells, hence inter-fiber space is their natural “niche”, and this is the reasons why we are currently focusing on such cell population, with the hope of favoring also artificial muscle vascularization.

Authors describe that the wet-spinning system was deeply characterized to investigate the relations between motor speed, fluid flowrates (bioink and calcium chloride solution) and fiber size. Can the data relating fluid flowrates to fiber size be shown ? (perhaps as supplementary information)

Authors' reply: We thank the Reviewer for this comment. Accordingly, we have introduced in the Expanded View of the revised manuscript version the Figure EV1 which contains all these data.

"After careful optimization, we decided to fabricate hydrogel fibers of around 100 μm ." Likely, authors mean 100 μm . Which parameter was optimized ?

Authors' reply: In that sentence, we indeed refer to the optimization of wet-spinning process. As mentioned above, in the revised version of the manuscript, we have introduced a new figure in the Expanded View (Figure EV1) in which we summarize such process aimed at identifying the best wet spinning conditions (i.e. the conditions that allow to obtain small hydrogel fibers of approx. 100 μm in diameter and a stable fiber production).

The size of the myo-substitute right after collection from the drum is not specified. What is the diameter of the drum ?

Authors' reply: This information has been introduced in the paragraph 2.3 *Biofabrication of 3D constructs*

From the drum, a ring structure is obtained. Such ring structure presumably is not what was implanted. How was the ring-shaped myo-substitute further treated ? Was there any *in vitro* culture of the myo-substitute before implantation? Was the whole ring structure implanted or a part of it ?

Authors' reply: After fabrication, samples were pre-cultured for 3 days *in vitro* and then implanted in TA. During the pre-culturing time, samples were kept as rings. At the moment of implantation, samples were sterile-cut with a scalpel to fill the lodge of the mouse TA which was previously removed.

If the latter, how much ? How does the ring-shape relate to the structure of the myo-substitute shown in Fig 4a (inset) ?

Authors' reply: As aforementioned, samples prior implantation were sterile cut to fit and fill the lodge of the removed TA. This means that only some part (random) of the fabricated and pre-cultured rings was ultimately implanted in the mice.

The myo-substitutes are described to contain aligned myotubes in culture after 2 weeks (P14). It would be good to guide the reader to fig 3 to support the claim. What were the culture conditions in which this was carried out ? Was any further shrinkage (after the initial 30%) of the myo-substitutes observed in vitro ?

Authors' reply: We thank the Reviewer for his/her comment. Accordingly, we have introduced the missing information in the paragraph 2.4 that has been renamed into 2.4 *In vitro cell and cell-laden sample culture*. As regards sample shrinkage, we did observe a partial compaction of the samples over the culturing time *in vitro*, however given the fiber/ring-shape of our samples, it was not straightforward a precise assessment. Based on samples length – i.e. ring length, a parameter easily measurable – samples' shrinkage at day 15 of *in vitro* culture was of approximately 10-15%. For the sake of clarity, we have introduced this information at the end of paragraph 3.3 *In vitro characterization of mouse-derived myo-substitutes*.

The human myo-substitutes seem to shrink dramatically between day 0 (Fig 8, second row, left) about 1 mm and day 3 (fig 8 second row second column) about 100 μm . Is this correct or is this some imaging artefact ?

Authors' reply: We thank the Reviewer for his/her comment that give us the opportunity to improve the clarity of our presented results. As regards the mentioned images in Figure 8, these two refer to a large part of a sample and to a single fiber respectively. In fact, as the Reviewer can possibly imagine, confocal imaging of such rings of fibers which are few mm in thickness and width is quite challenging and is often limited to few layers of fibers (200-300 μm in thickness). Specifically, the large view scan aimed at showing homogeneity throughout the sample while the single fiber scan aimed at showing myotube organization at the lowest level. In order to improve the clarity of this panel, we have introduced inset captions to explain that those images refer to a large view of a sample and to a single fiber.

From fig 2 it appear that the thickness of the myo-substitute is approximately 300-600 μm at day 0. This seems very thin as an implant. Also, the extruded ring is collected for 2 minutes at 50 rpm, which means 100 fibers of 100 μm thickness each; this would presumably be thicker ? In fig 2, only about 5 parallel fibers can be discerned. And in Fig S1, the myo-substitute seems even more thin, since 1 cell layer is relatively well in focus without much signal from cells above or below.

Authors' reply: One more time, we thank the Reviewer for his/her comment that give us the opportunity to improve the clarity of our presented results. As mentioned in the previous reply, bright field imaging of thick samples composed of tens of layers of fibers is challenging and most of the time only a very trained eye can discern cell organization in these samples. Therefore, to simplify cell organization recognition, samples presented in figure 2 were produced out of few layers of fibers. This missing information has been introduced in the caption of figure 2. For the sake of clarity, the same information has been introduced in the caption of Figure EV2.

The graft seems to expand in size quite considerably in vivo. From Fig 3 left image (in vitro) the size of the myo-substitute seems to be in the range of 1 mm. In Fig 4c, the thickness after 20 days in vivo seems to be around 4 mm. How can such increase be explained ? Is size of the graft perhaps in part due to oedema and/or macrophage cell infiltration ? In Fuoco et al. 2015, authors described that "even in an immunodeficient background, over time the xenogeneic graft with time attracted murine macrophages and other non-lymphoid cells that infiltrated and prevented myogenic maturation of the new muscle".

Authors' reply: We thanks the Reviewer for his/her observation. As mentioned in the previous reply, confocal imaging of such a complex structure is challenging and one is always limited to presenting only some parts of the samples, with the most stringent limit being in z-direction. In Figure 3 we have presented a large view (MIP) of a sample in which only fibers in-focus within 250 um along z-direction are shown, therefore, in this case, the overall dimensions of the presented scaffold should not be taken as the actual overall sample dimensions.

Nevertheless, what is comparable are the dimensions of the ablated TA portion and the myo-substitute obtained from part of the cell-laden yarn ring (Fig. 4a inset). As demonstrated by the histological sections, the reconstructed TA contain – besides Mabs – also macrophages, blood vessels, motoneurons and probably fibroblast, but no evident sign of oedema is visible.

From what mouse species were mouse Mabs derived ? Authors make reference to Fuoco et al 2012, but there the species was also not described. In the latter paper, authors reference Minasi et al 2002. These Mabs were derived from the dorsal aorta. Moreover, the animals used in Minasi were not SCID animals. Would it be possible that a xenogeneic response was mounted against the Mabs ?

Authors' reply: We apologize for the mistake. Mabs were isolated from C57/BL6 mice according to Diaz Manera et al., 2010 Cell Death Dis. Accordingly, we have introduced this information in the text. There is a slight xenogeneic response, despite using SCID/Beige mice, due to macrophage infiltration as shown in Figure EV3.

At what doubling were cells (human and mouse) used ? Can some basic characterization of the human cells be shown (eg % myogenic cells) ?

Authors' reply: We thank the Reviewer for his/her comment. Human cells have been already characterized in Vianello et al., Hum Mol Genet 2017. In our experiment, we have used human cells always at very low passage number (3 and 4), while Mabs have been used also at higher passages (9-10).

For the surgical procedure, was the TA completely ablated except for about 5% remaining on both ends attached to the tendon ? Or is the remaining 10 % still connecting the tendons ? It is unclear how the grafted myo-substitute is attaching to the remaining tendon. Especially given the force generated; 2N is considerable and it is intriguing how such force could be generated if not properly attached to the tendons. Characterization of the myotendinous junction would be of significant benefit to this study.

Authors' reply: We thank the Reviewer for this observation. The ablation was performed removing mainly the TA belly and leaving the remaining 10% connected with tendons. Thus, the implanted myo-substitute integrate with the remaining muscular tissue generating force through the host tendons. We have added this information in 2.5 *In vivo construct implantation* section. We agree with the Reviewer that myotendinous junction characterization would be interesting, but we believe that this issue is material for another paper.

It is impossible to discern a sarcomere structure in the areas designated with the asterisks in Fig 5.

Authors' reply: We have modified Figure 5 adding a higher magnification image showing clear sarcomeres in the LacZ+ myofibers.

A widespread laminin staining can be observed after implantation (Fig 5). This is very encouraging and may hint towards a very active extracellular matrix remodeling, almost

completely replacing the PEG-fibrinogen. However, it would be needed to include the necessary controls to exclude cross-reactivity/ background staining of the laminin antibody or autofluorescence of the used biomaterials.

Authors' reply: We respect Reviewer's opinion, but it is already quite clear from the presented images that Laminin signal surrounds MHC positive muscle fiber (Fig. 5 lower panel). Moreover, the commercial antibody against Laminin used from sigma is wide employed in all the laboratory working on skeletal muscle tissue, and we published several papers using the same antibody.

In the legend of Fig 5, authors state that myo-substitutes showed neo-vascularization, however, no markers of vascularization are presented in this figure.

Authors' reply: We thank the Reviewer for his/her comment. The mentioned issue was just a text layout problem. There was no neo-vascularization reference in figure 5.

Fig 6: what is the inset in panel b ?

Authors' reply: We thank the Reviewer for his/her comment. The inset in fig 6b was a slight magnification of a selected area of the figure 6 b itself. Accordingly, we have modified figure 6 by adding a new inset at higher magnification.

The methods section contains several sentences on the SMA quantification. However, it is not clear how the quantification of pixels translates to the quantification of number of vessels as presented in Figure 6.

Authors' reply: We apologize with the Reviewer for the missing information which we have now added in the *2.10 Vessel density evaluation* section. The vessel number has been evaluated by manual counting while pixels quantification was performed for vessel area and mean gray quantification.

The mere presence of SMA+ cells does not prove the presence of functional blood vessels. From supplementary figures 2 and 3, SMA positive cells can be seen, however a lumen can in most cases not be discerned. Moreover, the SMA may be derived from Mabs differentiated to smooth muscle cells. Supporting such interpretation is the co-localisation of vWF and LacZ staining on Fig 6c.

Authors' reply: We thank the Reviewer for his/her comment. As shown in Appendix Figures S2 and S3 (ex Supplementary Figure 2 and 3), one can clearly discern the vessel lumen in the native muscle tissue because vessels are mature and large. In the engrafted panels, instead, SMA signal labels smaller vessel as expected, but, albeit with less frequency, it is possible to appreciate the lumen of few larger vessels. Anyway, besides SMA we use a more selective staining - vWF - to quantify vascularization, and as shown in Figure 6d, one can notice that the SMA and vWF positive areas in the grafted TA are pretty the same. As stated before, Mabs are perivascular progenitor cells, hence the perivascular compartment is their natural “niche”, thus is normal to find LacZ positive nuclei around the vessel.

Why is there a large discrepancy between SMA and vWF staining in native muscle for number of vessels (Fig 6d) ? In contrast, in Fig 6e, it appears that more surface is SMA positive than vWF positive.

Authors' reply: We thank again the Reviewer for this observation. The discrepancy is related to the different behavior of SMA and vWF antibodies, the latter labeling also small capillaries missing smooth muscle layer. Therefore, a direct comparison of the measured numbers is not possible. This means that few SMA positive vessels can account in terms of area as many vWF positive vessels.

Fig 6c: As pointed out before, very few of the dystrophin+ areas (myofibers) seem to contain lacZ (transplanted cell) nuclei. However, (interestingly !) a number of green areas (vWF, endothelial cells) seem to be lacZ positive. This may indicate that part of the mesoangioblasts have differentiated to endothelial cells. This would warrant further investigation and if true, may provide a positive rationale to the use of mesoangioblasts.

Authors' reply: As stated above, Mabs are perivascular progenitor cells and their capacity in differentiating also towards additional cell lines is one of the main reasons why we are now investigating these cells for muscle regeneration.

Fig6d is a quantification of the number of vessels per 0.2 mm². This area corresponds more or less to the image area shown in Fig 6b (0.5x0.4 mm) as judging from the scalebar in panel a). It seems a bit of a stretch to identify 40 vessels in this figure. It rather seems that only 1 vessel-like structure (with an identifiable lumen) can be observed.

Authors' reply: Respecting Reviewer's opinion, we would like to stress the point that around each myofiber there are a range of 2-6 capillary, hence the number of vessels scored is not

overestimated. Looking at new figure 6b inset at higher magnification (150x150 μm), one can see several vWF positive capillary characterized by a very small, unmeasurable lumen.

In Fig 7b, it seems that the areas where neuF is observed are relatively far apart from a lacZ nucleus. Therefore, wouldn't this be indicative of host-derived myofibers which are in proximity to axons (NeuF staining) ?

Authors' reply: We thank the Reviewer for his/her comment. Accordingly, we have introduced a new figure 7c highlighting the innervation of the numerous LacZ positive myofibers.

In the functional test (Fig 7d), the data for a native muscle are shown as a grey zone. It is unclear what this zone is based on. To compare data and as a positive control, it would be needed to show actual measurements for the native situation with the used experimental setup.

Authors' reply: We appreciate Reviewer's comment, nevertheless this setup has been used in several publication evaluating wt TA force showing always the same values. In figure 7d, we indicated only the tetanic force for the sake of clarity, anyway we inserted in the appendix a new table with the wt strength values.

Fig 8: please label panels. In the figure of the bulk hydrogel, second to the left, it seems that many nuclei are MHC negative. Does this mean that the used cells are to a large extent non-myogenic ?

Authors' reply: We thank the Reviewer for the comment and accordingly we modify image labelling. First of all we would like to emphasize that all the immunofluorescence figures are presented as MIP given the 3D nature of our samples. We believe that the main explanation to this issue is related to optical problems, in particular out of focus issues. In fact, DAPI signal is generally much sharper and strong than those obtained for secondary antibodies. In simpler words, it is quite common to observe in 3D structures DAPI signal from out-of-focus layers also in the focused one. Nevertheless, we cannot exclude at this moment the possibility of a presence of non-myogenic cells in the pool of cells embedded in the constructs.

Fig S1 : please provide scale bars.

Authors' reply: Scale bars have been introduced in the new Figure EV2.

Word errors:

P1 The sentence starting with "Of note ..."is grammatically not correct

P2: a reliable...models -> model

Consistently use either mesoangioblasts or mesangioblasts

P6: isolated and culture -> isolated and cultured

P11: $P < 0.05$ -> $P < 0.05$

P11: unprecedent -> unprecedented

P12 easier to use then -> easier to use than

P15: we have not leaved out -> we have not left out

Authors' reply: We thank the Reviewer for the corrections and accordingly we have modified the text.

22nd Dec 2020

Dear Dr. Gargioli,

Thank you for the submission of your revised manuscript to EMBO Molecular Medicine. I am pleased to inform you that we will be able to accept your manuscript pending the following final amendments:

1) Please address all the points raised by the referee #3.

2) Figures:

- Please upload separate files for each EV figure.
- Remove EV figure legends from the figure files and include them into the main manuscript file after the main figure legends.
- Use upper case letters to label the figure panels.

3) In the main manuscript file, please do the following:

- Correct/answer the track changes suggested by our data editors by working from the attached/ uploaded document.

***** Reviewer's comments *****

Referee #1 (Remarks for Author):

Although some point remains to be fully elucidated, the authors adequately and sufficiently addressed the points I raised

Referee #3 (Remarks for Author):

The authors did a good job with the revision. Only a minor issue needs some attention: In the assessment of vascularization, there seems to be a discrepancy between the data presented in Figure 6e versus Figure S3. In Figure 6e, the SMA-positive area (light grey bars) of graft is higher than native. In Figure S3 which is also a SMA quantification, vessel area of graft is about half of native and this is indicated as statistically significant. Can this be clarified ?

Furthermore, just 2 typographical suggestions:

Page 33: Figure EV56 is likely EV5

Legend of Figure 6d: vassels->vessels

The authors performed the requested editorial changes.

Point by point letter

Referee #3 (Remarks for Author):

The authors did a good job with the revision. Only a minor issue needs some attention:
In the assessment of vascularization, there seems to be a discrepancy between the data presented in Figure 6e versus Figure S3. In Figure 6e, the SMA-positive area (light grey bars) of graft is higher than native. In Figure S3 which is also a SMA quantification, vessel area of graft is about half of native and this is indicated as statistically significant. Can this be clarified?

Authors' reply: We appreciate Reviewer's comment and we apologize for the oversight. We analyzed the vascularization employing different ROI set up, but the more appropriate was the last used, with a larger ROI, showed in the figure 6 revealing a more reliable analysis used also for vWB quantification. While in the supplementary was left by mistake an old chart, indeed the two charts displayed different dimension range demonstrating a different analysis employing a smaller ROI. Hence, we modified Appendix Table S3.

Furthermore, just 2 typographical suggestions:

Page 33: Figure EV56 is likely EV5

Legend of Figure 6d: vassels->vessels

Authors' reply: We thank the Reviewer for his/her advice, we corrected the mistake.

12th Jan 2021

Dear Dr. Gargioli,

We are pleased to inform you that your manuscript is accepted for publication.

Corresponding Author Name: Cesare Gargioli
Journal Submitted to: EMBO Molecular Medicine
Manuscript Number: EMM-2020-12778